# Direct oxidative carbonylation of methane to acetic acid via high-valent iron-oxo mediated water activation

Haonan Zhang[1,7], Richard J. Lewis [2,7], A. Iulian Dugulan [3,7], Yang Li[1], Shuai Wang[1], Zhenxing Wang [4], Jianrong Zeng[5], Nicholas F. Dummer[2], Yanyan Xi[1], Yunyun Li[1], Thomas E. Davies[2], Mingbo Wu [1,6] ✉, Graham J. Hutchings [2] ✉ & Wenting Wu [1] ✉

Direct conversion of $CH_4$ into value-added chemicals is impeded by the inert C-H bonds and inefficient C-C coupling. We report a spatially separated Rh-O-Fe active-site architecture that decouples $CH_4$ and $H_2O$ activation through a high-valent-metal mediated radical mechanism, enabling selective $CH_3COOH$ synthesis. In-situ infrared, operando Mössbauer spectroscopy, and quasi in-situ high-field EPR reveal that $O_2$ oxidizes Rh and Fe to high valence states. $Rh^{(III)}$ activates $CH_4$ to •$CH_3$, while $Fe^{(IV)} = O$ dissociates $H_2O$ into •OH through a truncated water-gas shift pathway. •OH rapidly reacts with CO to form •COOH intermediates, which couples with •$CH_3$ within the zeolite to yield $CH_3COOH$. This dual-site strategy circumvents kinetic limits of conventional water-gas shift and CO insertion steps. The catalyst achieves 18.2 mmol $g_{cat}^{-1}$ $h^{-1}$ $CH_3COOH$ with 92% selectivity and 100-hour stability in continuous operation. This study establishes radical decoupling enabled by high-valent metal sites as a design principle for selective alkane oxidation.

Methane ($CH_4$), the primary component of natural gas, stands as a critical yet underutilized carbon resource due to the formidable challenges in its selective activation and functionalization[1–3]. The direct conversion of $CH_4$ into $C_{2+}$ products demand overcoming two intrinsic barriers: the high bond dissociation energy of $sp^3$ C-H bonds (439 kJ mol⁻¹) and the sluggish kinetics of C-C bond formation[4–6]. Acetic acid ($CH_3COOH$), a cornerstone chemical with an annual demand of 18 million metric tons[7,8], exemplifies this challenge, as its industrial production still relies on an indirect syngas-derived methanol carbonylation process plagued by energy intensity[9–12]. The one-step methane oxidation process for $CH_3COOH$ production enables

efficient synthesis through direct $CH_4$ activation under mild conditions, bypassing the complex steps required in traditional methanol carbonylation processes, such as methanol synthesis and the use of halogen promoters, thus demonstrating significant advantages in feedstock economy and process simplicity. Although this new pathway still requires further refinement, it has already provided a more atom efficient and sustainable alternative for the green production of $CH_3COOH$.

Current strategies for direct $CH_4$ oxidative still face several critical limitations. For C-H activation, conventional strategies typically rely on unstable zero-valent metals (e.g., $Rh^0$) or external addition of $H_2O_2$ to

[1]State Key Laboratory of Heavy Oil Processing, College of Chemistry and Chemical Engineering, China University of Petroleum (East China), Qingdao, PR China. [2]Max Planck-Cardiff Centre on the Fundamentals of Heterogeneous Catalysis FUNCAT, Cardiff Catalysis Institute, School of Chemistry, Cardiff University, Cardiff, UK. [3]Fundamental Aspects of Materials and Energy (FAME), Department of Radiation Science and Technology (RST), Delft University of Technology, Delft, the Netherlands. [4]Wuhan National High Magnetic Field Center, Huazhong University of Science and Technology, Wuhan, China. [5]Shanghai Synchrotron Radiation Facility, Shanghai Advanced Research Institute, Chinese Academy of Sciences, Shanghai, China. [6]State Key Laboratory of Advanced Optical Polymer and Manufacturing Technology, College of Chemical Engineering, Qingdao University of Science & Technology, Qingdao, PR China. [7]These authors contributed equally: Haonan Zhang, Richard J. Lewis, A. Iulian Dugulan. ✉e-mail: wumb@upc.edu.cn; hutch@cardiff.ac.uk; wuwt@upc.edu.cn

**Table 1 | Catalytic performance of RhFe/ZSM-5 catalysts for the oxidation of $CH_4$**

| $CH_4+O_2+CO\rightarrow CH_3COOH$ | | | | | | | | | | |
|---|---|---|---|---|---|---|---|---|---|---|
| Entry | Catalyst | T (K) | Reactants (MPa) | | | Productivity (mmol $g_{cat}^{-1}$ $h^{-1}$) | | | $CH_3COOH$ Selectivity (%)[d] |
| | | | $CH_4$ | $O_2$ | CO | $CH_3OH$ | HCOOH | $CH_3COOH$ | |
| 1 | RhFe/ZSM-5 | 463 | 3 | 0.3 | 0.6 | 0.15 | 1.36 | 18.27 | 92 |
| 2 | RhFe/ZSM-5 | 363 | 3 | 0.3 | 0.6 | 0.02 | 0.03 | 0.17 | 77 |
| 3 | RhFe/ZSM-5 | 463 | 0 | 0.3 | 0.6 | n.d.c | n.d.c | n.d.c | n.d.c |
| 4 | RhFe/ZSM-5 | 463 | 3 | 0 | 0.6 | n.d.c | n.d.c | n.d.c | n.d.c |
| 5 | RhFe/ZSM-5 | 463 | 3 | 0.3 | 0 | n.d.[c] | n.d.[c] | n.d.[c] | n.d.[c] |
| 6 | H-ZSM-5 | 463 | 3 | 0.3 | 0.6 | n.d.[c] | n.d.[c] | n.d.[c] | n.d.[c] |
| 7 | Rh/ZSM-5 | 463 | 3 | 0.3 | 0.6 | 0.68 | 1.34 | 3.2 | 61 |
| 8 | Fe/ZSM-5 | 463 | 3 | 0.3 | 0.6 | n.d.[c] | n.d.[c] | n.d.[c] | n.d.[c] |
| 9[a] | Rh/ZSM-5//Fe/ZSM-5 | 463 | 3 | 0.3 | 0.6 | 0.71 | 1.42 | 3.64 | 63 |
| 10[b] | Rh/ZSM-5-C | 463 | 3 | 0.3 | 0.6 | 0.4 | 1.15 | 1.89 | 55 |
| 11[b] | Fe/ZSM-5-C | 463 | 3 | 0.3 | 0.6 | n.d.[c] | n.d.[c] | n.d.[c] | n.d.[c] |
| 12[b] | RhFe/ZSM-5-C | 463 | 3 | 0.3 | 0.6 | 3.8 | 1.58 | 5.08 | 49 |

Reaction conditions: catalyst (10 mg), water (20 mL), time (2 h), stirring speed 800 rotations per minute (rpm). For entries 3–5, the total pressure was maintained with $N_2$ or Ar.
[a]Rh/ZSM-5 and Fe/ZSM-5 were physically mixed.
[b]ZSM-5-C was obtained from Nankai University Catalyst Co., Ltd., which has the same metal loading capacity and $SiO_2$: $Al_2O_3$ ratio (18) as RhFe/ZSM-5.
[c]*n.d.* not detected.
[d]$CH_3COOH$ selectivity in liquid products.

generate •$OH^{13}$. In some cases, the complete water-gas shift (WGS) cycle ($CO + H_2O \rightarrow CO_2 + H_2$) is involved to in-situ produce $H_2O_2^{14-16}$, which then undergoes multiple steps to form •OH for $CH_4$ activation. This multistep pathway not only prolongs reaction process but with progressively declining energy efficiency, significantly limiting the overall catalytic performance. For C-C coupling with CO, a kinetically sluggish process remains the rate-determining bottleneck. For example, the methanol carbonylation proceeds efficiently through a migratory insertion of the neutral covalent ligands •$CH_3$ and *CO at a metal center, facilitated by $CH_3I$, enabling high-efficiency carbonylation[17]. Therefore, in the direct $CH_4$ carbonylation, the in-situ generated •$CH_3$ do not readily couple with neutral $CO^{18}$. Although recent efforts have focused on enhancing •OH and •$CH_3$ concentrations or optimizing active sites[19,20], the critical C-C coupling step (involving •$CH_3$-CO vs. •$CH_3$-•COOH) in this pathway has never been experimentally verified, hindering the development of efficient catalytic systems.

To overcome the kinetic and mechanistic constraints of direct $CH_4$ carbonylation, we report a catalyst design that integrates C-H activation with spatially orchestrated radical coupling. By constructing Rh-O-Fe active sites within ZSM-5, the RhFe/ZSM-5 catalyst leverages a CO-assisted water activation mechanism that circumvents the inefficiencies of conventional WGS cycles. Here, $Rh^{(III)}$ can active $CH_4$ to generate •$CH_3$, while $O_2$ treatment sustains the dynamic valence transition of $Rh^{(III)}$, and simultaneously oxidizes Fe sites to form $Fe^{(IV)} = O$ species. This high-valent Fe-oxo entity facilitates direct cleavage of $H_2O$ into •OH, which rapidly reacts with CO to produce •COOH intermediates. These subsequently couple with Rh-derived •$CH_3$ species within the confined micropores of the zeolite to form $CH_3COOH$. This integrated mechanism bridges the long-standing gap between $CH_4$ activation and selective C-C bond coupling, providing a new paradigm for efficient and scalable natural gas valorization.

## Results
### Catalytic performance of $CH_4$ carbonylation into acetic acid
Monometallic ZSM-5-based catalysts with 0.6 wt% metal loading (Rh, Fe, Ni, Co, Cu, Zn, Ag, Pd, Au, Re, Pt, Mo) were synthesized via a template-free seeded growth method and evaluated in a batch reactor. Among these, Rh/ZSM-5 exhibited the highest catalytic performance, achieving a $CH_3COOH$ yield of 3.2 mmol $g_{cat}^{-1}$ $h^{-1}$ with a selectivity of

61% at 463 K, as determined by GC and $^1H$ NMR (Supplementary Fig. 1-2). Notably, the Fe monometallic has little reactivity. In contrast, introducing Fe into Rh/ZSM-5 to form RhFe/ZSM-5 significantly improved catalytic performance, yielding 18.2 mmol $g_{cat}^{-1}$ $h^{-1}$ of $CH_3COOH$ with a selectivity of 91.8% at 463 K under optimized reaction conditions (Table 1 entry 1, Supplementary Figs. 3, 4). The corresponding TOF reached ~216 $h^{-1}$, which is ~2.5 times that of monometallic Rh/ZSM-5 (92 $h^{-1}$) (Supplementary Fig. 5). It is worth mentioning that the reaction was initiated at 363 K (Table 1 entry 2). This performance was significantly higher than AuFe/ZSM-5, PdFe/ZSM-5, RuFe/ZSM-5, PtFe/ZSM-5, RhNi/ZSM-5, RhCu/ZSM-5, and different supports, as well as previously reported catalysts in the literature (Fig. 1a, Supplementary Fig. 6, 7 and Supplementary Table 1).

Furthermore, when Rh/ZSM-5 and Fe/ZSM-5 catalysts were physically mixed and tested under identical reaction conditions, no significant improvement in $CH_3COOH$ yield or selectivity was observed compared with Rh/ZSM-5 (Supplementary Fig. 2b), highlighting the crucial role of the synergistic interaction between Rh and Fe in RhFe/ZSM-5. The lower activity of the spatially separated RhFe/ZSM-I catalyst further confirms that Rh-Fe synergy is essential for high performance. Highly active Rh-Fe sites not only enhance $CH_4$'s C-H bond activation but also modify the reaction pathway of $CH_4$ oxidative coupling. Additionally, a systematic investigation of Fe content, while maintaining Rh loading at 0.6 wt.%, revealed a gradual increase in both $CH_3COOH$ yield and selectivity (Supplementary Fig. 8), further demonstrating that the Fe introduction plays a key role in optimizing the reaction mechanism.

For comparison, a catalyst was synthesized using a commercial ZSM-5 (Si:Al=18) support via an impregnation method (denoted as RhFe/ZSM-5-C), and exhibited a significantly lower $CH_3COOH$ yields of 5.08 mmol $g_{cat}^{-1}$ $h^{-1}$ with a selectivity of 55.5% (Supplementary Fig. 9). BET surface area analysis revealed similar values for ZSM-5-C (373 m2/g) and ZSM-5 (338 $m^2/g$), indicating that the differences in catalytic performance stem primarily from variations in metal active sites or acidity rather than textural properties (Supplementary Fig. 10).

$CH_3COOH$ formation was observed only when $CH_4$, CO, and $O_2$ were simultaneously present, as no detectable products were detected in control experiments omitting any of these components (Table 1, Entry 3−5). These findings demonstrate that $CH_4$, CO, $O_2$, and $H_2O$ are all essential for $CH_3COOH$ production. Additionally, when $H_2$ was used

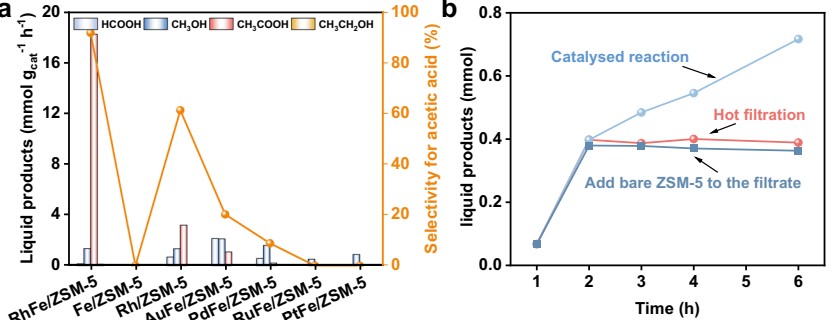
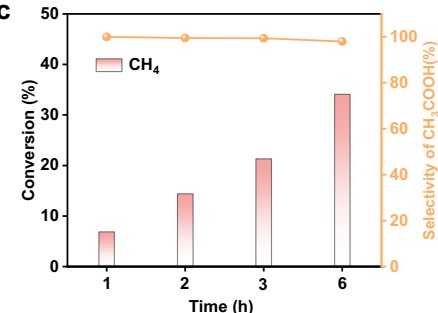

**Fig. 1 | Catalytic performance evaluation of CH₄ oxidative coupling reaction.** **a** Catalytic performance with different metal over ZSM-5, reaction conditions: 463 K, 10 mg catalyst, 2 h, 20 mL $H_2O$, 30 bar $CH_4$, 3 bar $O_2$, 6 bar CO, and stirring speed 800 rotations per minute (rpm). **b** Time-dependent conversion and hot filtration test of the reaction. **c** Oxygenate yield and selectivity as a function of $CH_4$ conversion at lower $CH_4$ partial pressure, reaction conditions: 463 K, 10 mg catalyst, 20 mL $H_2O$, 121 umol $CH_4$, 11 bar Air, 6 bar CO, 18 bar $N_2$, and stirring speed 800 rotations per minute (rpm).

as a substitute for CO, only $C_1$ oxygenates, such as $CH_3OH$, HCOOH, were detected (Supplementary Fig. 11a and Supplementary Table 2 Entry 1). This suggests that $CH_3COOH$ is produced through the C-C coupling of $CH_4$ and CO, rather than from environmental contaminants or the catalyst itself. In addition, comparative experiments performed with $CH_3COOH$ in the presence and absence of CO confirmed that $CO_2$ is predominantly formed through CO oxidation (Supplementary Fig. 11b). Under standard methane oxidation conditions, the rate of $CO_2$ production reached 41.6 mmol $g_{cat}^{-1}$ $h^{-1}$, surpassing that from $CH_3COOH$. For clarity, both liquid-phase selectivity (>92% toward $CH_3COOH$ among oxygenated products) and total carbon selectivity (including $CO_2$, 42.3% for $CH_3COOH$) are provided (Supplementary Figs. 12, 13). These results indicate that, although the catalyst shows high selectivity to acetic acid in the liquid phase, the oxidation of CO to $CO_2$ represents a major competing pathway and warrants further optimization in future studies.

To confirm the heterogeneous nature and stability of the catalyst, further control experiments were conducted[21]. In the absence of catalyst, as well as in the presence of $FeCl_2$, $Rh(NO_3)_3$, or a mixture of these species, no catalytic activity was observed (Supplementary Table 2 Entry 2–5). Hot filtration experiments further confirmed the heterogeneous nature of the catalytic process, as no additional reaction occurred after 2 h (Fig. 1b). The stability of RhFe/ZSM-5 was evaluated over five reaction cycles, showing negligible leaching of Rh and Fe species, as determined by ICP analysis (Supplementary Table 3). In addition, batch reactions conducted under lower $CH_4$ partial pressure in air and CO achieved a $CH_4$ conversion of up to 24.8% within 3 h (Fig. 1c), with > 98% selectivity to $CH_3COOH$. Even when prolonging reaction time to 6 h, no methanol and formic acid were not detected by ¹H-NMR (Supplementary Fig. 14). Furthermore, this reaction system is effective not only for $CH_4$ but also for ethane, which yield acids with an additional carbon atom (Supplementary Fig. 15).

## Identification of active sites

The synergy observed by the presence of both Rh and Fe in the $CH_4$ reaction to $CH_3COOH$ over the prepared ZSM-5-based catalyst required careful study to elucidate a structure-activity relationship. XRD analysis shows that the ZSM-5 crystal structure was well preserved in Fe/ZSM-5, Rh/ZSM-5, and RhFe/ZSM-5[22], with no characteristic diffraction peaks corresponding to Rh and Fe metals (Supplementary Fig. 16). The isolated Fe and $RhO_X$ nanoparticles species were detected in the ZSM-5 as confirmed by the annular dark-field scanning transmission electron microscopy (AC-HAADF-STEM) and EDX (Fig. 2a, b and Supplementary Figs. 17, 18). In the UV-vis spectrum (Supplementary Fig. 19), no obvious absorption at >400 nm was observed, which implies that homogeneous isolated Fe species were confined in the micropores[3,23].

X-ray absorption fine structure spectroscopy (XAS) was used to systematically study electronic structures and coordination environments of as-prepared catalysts. In the Fourier transform (FT) $k^3$-weighted extended EXAFS spectrum, the absence of Fe-Fe shell verifies that the Fe atoms were atomically dispersed in the Fe/ZSM-5 and RhFe/ZSM-5 (Fig. 2c). Rh exhibits a $RhO_X$ curve similar to that of $Rh_2O_3$ (Fig. 2d). For Rh/ZSM-5, the best fitting of the Rh-O backscattering path reveals the Rh-O coordination numbers of 6 in the first coordination shell (Supplementary Fig. 20, Supplementary Table 4). For Fe/ZSM-5, the major peak at ~1.50 Å in Fe K-edge EXAFS spectrum could be attributed to the Fe-O configurations with a coordination number of 4. For RhFe/ZSM-5, a primary peak at 1.52 Å could be attributed to Rh-O shell, and Rh-O-Fe in second shell in 2.42 Å was observed (Fig. 2d), which is longer than that of Fe-O-Fe in Fe/ZSM-5 (2.34 Å), which suggests the presence of an Rh-O-Fe structure in RhFe/ZSM-5.

To confirm the Rh-O-Fe structure, ⁵⁷Fe Mössbauer spectra of Fe/ZSM-5 and RhFe/ZSM-5 were collected at 4.2 K using a ⁵⁷Co(Rh) source (Fig. 2e, f and Supplementary Table 5). The spectra were fitted with a doublet (assigned to dimeric high-spin $Fe^{(III)}$-$Fe^{(III)}$ species) and a broad sextet (assigned to isolated $Fe^{(III)}$ ions). Minor isolated $Fe^{(II)}$ species (2–3%) were also detected. The $Fe^{(III)}$ paramagnetic doublet at 4.2 K indicates well-distributed Fe species without sufficient neighboring Fe atoms to generate a magnetic field, even at cryogenic temperatures. The doublet suggests spin relaxation (spin-flipping) due to coupling between at least two neighboring Fe atoms, supporting a dimeric structure. When $Fe^{(III)}$ ions are spaced >1.5 nm apart, slow spin-spin relaxation results in paramagnetic hyperfine splitting. The broad sextets correspond to isolated $Fe^{(III)}$ ions with large hyperfine fields (>55 T), as shown in Supplementary Fig. 21, with a maximum magnetic field distribution at -56 T. Typical Fe oxide clusters (-54 T) are likely absent. Paramagnetic hyperfine splitting in Fe-ZSM-5 aligns with prior observations[24]. Isomer shifts for $Fe^{(III)}$ species suggest octahedral extra-framework coordination, as tetrahedral positions would exhibit smaller shifts (-0.4 mm/s). The Rh-Fe sample shows more dimeric species (28%) than the Fe-only sample (18%), potentially indicating Fe-Rh dimer formation, consistent with XAS results.

We further investigated the interaction between Fe and Rh through electronic structure analysis. The Rh K-edge XANES spectrum (Supplementary Fig. 22a) showed that the introduction of Fe into Rh/ZSM-5 increased the oxidation state of Rh in RhFe/ZSM-5 when compared to that in Rh/ZSM-5. Similarly, Fe K-edge XANES spectrum (Supplementary Fig. 22b) revealed that Fe in RhFe/ZSM-5 has a lower oxidation state than in Fe/ZSM-5, indicating electron transfer from Rh to Fe. The oxidation state of Fe is between +2 and +3, while Rh is below +3. This charge redistribution was further supported by the Rh 3$d$ XPS spectrum (Supplementary Fig. 23), where the characteristic peaks at 312.5 eV and 307.7 eV in Rh/ZSM-5 shift to 312.9 eV and 308.2 eV in

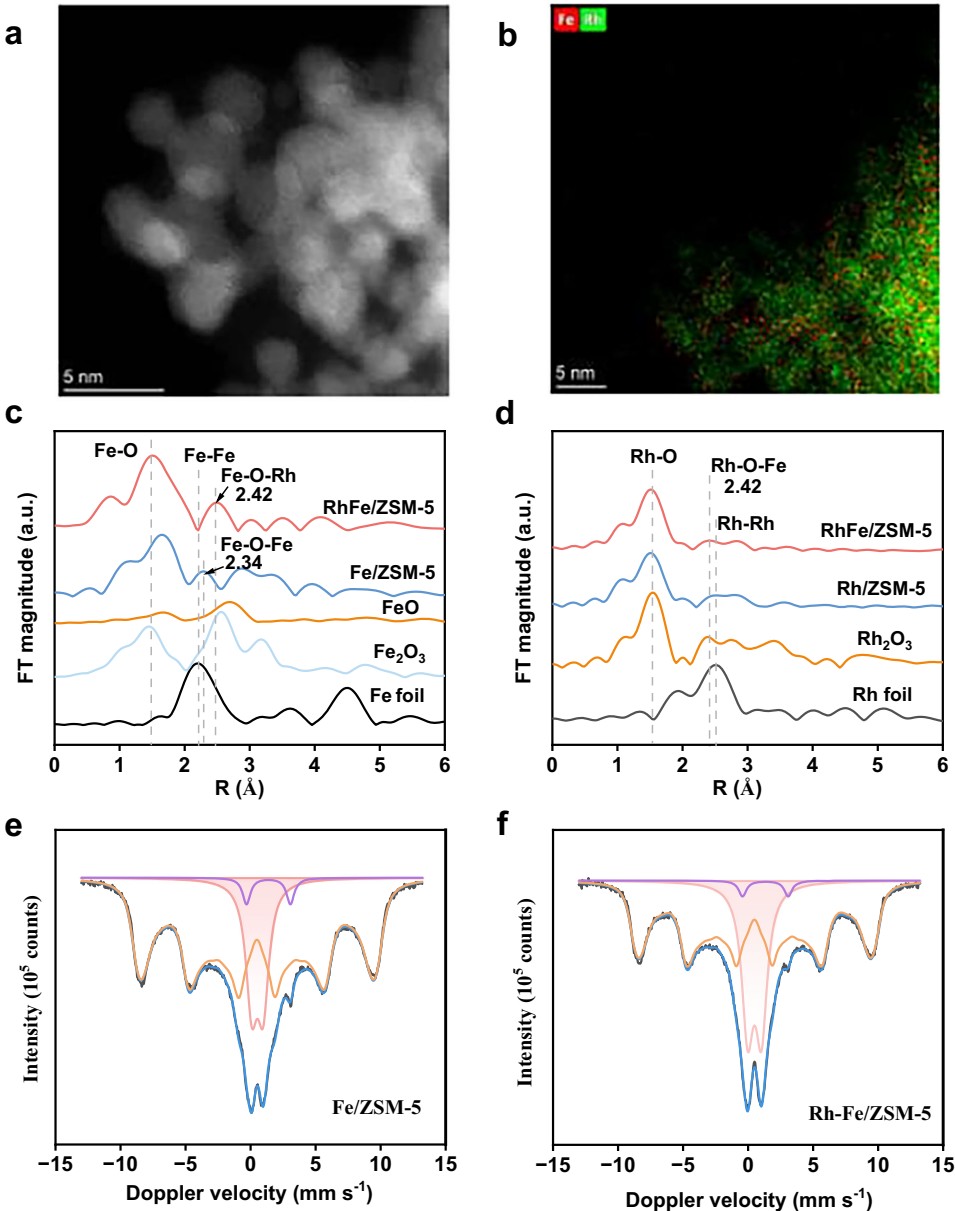

**Fig. 2 | Structural characterization. a** AC-HAADF-STEM image of RhFe/ZSM-5. **b** HAADF-STEM-EDS mapping of the RhFe/ZSM-5. **c** Fourier transform (FT) $k^3$-weighted EXAFS spectra of the Fe/ZSM-5, RhFe/ZSM-5, FeO, Fe$_2$O$_3$, and Fe foil.

**d** Fourier transform (FT) $k^3$-weighted EXAFS spectra of the Rh/ZSM-5, RhFe/ZSM-5, Rh$_2$O$_3$ and Rh foil. **e, f** $^{57}$Fe Mössbauer spectra for the Fe/ZSM-5 and RhFe/ZSM-5 catalysts.

RhFe/ZSM-5, suggesting an increased electron density on Fe in RhFe/ZSM-5.

To further elucidate the unique electronic structure of the catalyst, we carried out in-situ CO-DRIFTS measurements (Supplementary Fig. 24). ZSM-5 and monometallic Fe/ZSM-5 show characteristic peaks of gaseous CO at 2171 cm$^{-1}$ and 2118 cm$^{-1}$, whereas Rh/ZSM-5 exhibits linear CO adsorption bands at 2107 cm$^{-1}$ and 2040 cm$^{-1}$. Notably, RhFe/ZSM-5 catalyst shows a distinct blue shift to 2104 cm$^{-1}$ and 2037 cm$^{-1}$, indicating a decrease in the electron density of Rh sites. This directly reflects the electronic interaction between Rh and Fe, specifically, electron transfer from Rh to Fe, and is fully consistent with XPS and XANES data. In addition, the appearance of a bridged CO adsorption peak at 1860 cm$^{-1}$ further confirms the presence of Rh nanoparticles, in agreement with HAADF-STEM observations.

To directly investigate the influence of acidic sites on catalytic performance, we prepared a series of RhFe/ZSM-5-X catalysts with identical metal loadings but varying SiO$_2$/Al$_2$O$_3$ ratios (X = 18, 25, 40,

60, 80). NH$_3$-TPD and pyridine adsorption infrared spectroscopy (Py-IR) results show that the acidity of the catalysts significantly decreases with increasing SiO$_2$/Al$_2$O$_3$ ratios (Supplementary Fig. 25a, b). Catalytic performance tests (Supplementary Fig. 25c) indicate that both the yield and selectivity of CH$_3$COOH decrease markedly as the support SiO$_2$/Al$_2$O$_3$ increases. Among these, RhFe/ZSM-5-18 exhibits the best CH$_3$COOH performance (18.2 mmol g$_{cat}^{-1}$ h$^{-1}$), while RhFe/ZSM-5-80 shows a substantial decline in activity (0.4 mmol g$_{cat}^{-1}$ h$^{-1}$). This clearly demonstrates that zeolite acidity is a key factor in maintaining high catalytic performance.

To further differentiate the acidic contribution, Na$^+$ ion exchange was performed on RhFe/ZSM-5 (SiO$_2$/Al$_2$O$_3$ = 18) to obtain RhFe/Na-ZSM-5. Py-IR reveals a significant decrease in both Brønsted and Lewis acid sites, confirming effective neutralization of the acidic sites (Supplementary Fig. 26a). Correspondingly, the CH$_3$COOH yield over RhFe/Na-ZSM-5 decreases to 3.8 mmol g$_{cat}^{-1}$ h$^{-1}$, further proving that acidic sites (particularly exchangeable H$^+$) are indispensable for high catalytic

activity (Supplementary Fig. 26b). In CO-TPD (Supplementary Fig. 26c), RhFe/ZSM-5 exhibits a strong CO desorption peak at 665 K, indicating its strong adsorption capacity for CO. In contrast, the desorption peak for RhFe/Na-ZSM-5 shifts to 553 K with reduced intensity, demonstrating that Na$^+$ exchange not only weakens acidity but also the CO adsorption strength of catalyst.

## Comparison with conventional WGS and key steps

Previous studies have proposed that water in the Water-Gas Shift (WGS) reaction plays a crucial role in CH$_4$ activation[14,25], as H$_2$ generated from WGS facilitates the conversion of O$_2$ into H$_2$O$_2$, which subsequently produces •OH species for CH$_4$ activation. However, this multi-step process introduces significant kinetic limitations that hinder overall reaction efficiency. In contrast, our RhFe/ZSM-5 catalytic system appears to follow a fundamentally different pathway.

Isotope kinetic studies in the WGS system (CO and H$_2$O) catalyzed by RhFe/ZSM-5 showed a kinetic isotope effect (KIE) ratio of $K_H/K_D$ (H$_2$) = 4 at 463 K (Supplementary Fig. 27), confirming that H$_2$O decomposition is the rate-determining step (RDS) in the WGS reaction. However, in the CH$_4$, CO, O$_2$, and H$_2$O system, the KIE ratio for CH$_3$COOH formation over Rh/ZSM-5 was approximately 1 (Fig. 3a and Supplementary Fig. 28), whereas RhFe/ZSM-5 exhibited an inverse KIE ($K_H/K_D$ = 0.21) (Fig. 3b and Supplementary Figs. 29, 30), suggesting that the presence of Fe promotes H$_2$O activation, thereby eliminating water activation as the rate-determining step in the reaction.

Although a low concentration of H$_2$ was detected during WGS over RhFe/ZSM-5 (Supplementary Fig. 31), the significant difference between the formation rates of H$_2$ and liquid products suggests that H$_2$ does not play a major role in CH$_3$COOH formation. This conclusion is further corroborated by the effect of H$_2$ addition on product selectivity, as the introduction of H$_2$ resulted in only a 1.5 increase in CH$_3$COOH yield, whereas methanol yield increased by 3.7 (Supplementary Fig. 32). In the in-situ DRIFTS measurements of CO, O$_2$, and H$_2$O reaction system (Fig. 3c, d), two obvious signals at 950 cm$^{-1}$ and 932 cm$^{-1}$ were observed on Rh/ZSM-5 and RhFe/ZSM-5, which can be assigned to side-on O$_2$ adsorption and O-O stretching of surface peroxo species[26–28]. Moreover, a new peak at 840 cm$^{-1}$ and a shoulder at 864 cm$^{-1}$ were detected on Rh/ZSM-5 (Fig. 3c), attributed to the O-O

stretching of adsorbed and free H$_2$O$_2$ molecules, respectively[28]. Nevertheless, these two signals were absent on RhFe/ZSM-5 (Fig. 3d), indicating that H$_2$O$_2$ wasn't generated on RhFe/ZSM-5.

## Identification of active species and oxygen sources

In order to study the influence of •OH, excess H$_2$O$_2$ instead of O$_2$ was added into the system of CH$_4$, CO, and H$_2$O (Supplementary Fig. 33). Both the yield and selectivity of CH$_3$COOH were lower than those observed in the CH$_4$, CO, O$_2$, and H$_2$O reaction system. Notably, although the concentration of •OH radicals in the H$_2$O$_2$ system was significantly higher than in the CO and O$_2$ systems (Supplementary Fig. 34), the CH$_3$COOH yield remained relatively low, indicating that •OH radicals can participate in CH$_3$COOH formation, but not the primary pathway of CH$_4$ activation. These findings demonstrate that an alternative activation mechanism must be responsible for CH$_4$ conversion rather than H$_2$ + O$_2$ → H$_2$O$_2$ → •OH pathway.

In the isotope-labeling experiments of CH$_4$, CO, $^{16}$O$_2$, and H$_2$$^{18}$O (Fig. 3e, Supplementary Fig. 35), isotopically labeled CH$_3$CO$^{18}$OH was detected, whereas only CH$_3$CO$^{16}$OH was observed when CH$_4$, CO, $^{18}$O$_2$, and H$_2$$^{16}$O were used. These results confirm that the hydroxyl oxygen in CH$_3$COOH originates from H$_2$O decomposition rather than O$_2$. Additionally, when using CH$_4$, CO, O$_2$, and D$_2$O as reactants, CH$_3$COOD was detected, demonstrating that the hydrogen in the hydroxyl group of CH$_3$COOH originates from H$_2$O.

## Reaction intermediates and coupling mechanism

To explore the intermediates after introducing Fe active sites in this reaction system, we carried out in-situ Mössbauer spectroscopy analysis[29] (Fig. 4a, b and Supplementary Tables 6, 7) and high-frequency (240 Hz) quasi in-situ electron paramagnetic resonance (HF-EPR) spectroscopy analysis (Fig. 4c)[30,31]. In the in-situ Mössbauer measurements, both the source and the samples were cooled to 4.2 K after undergoing reaction at 463 K. In this way, the data can directly be compared to data obtained at room temperature, since both source and absorbers were maintained at equal temperatures, thereby experiencing similar second-order Doppler shifts.

After an initial O$_2$/Ar treatment (Fig. 4a, b, Treatment A), the presence of Fe$^{(IV)}$ = O species were clearly in the sub-Mössbauer-spectra of

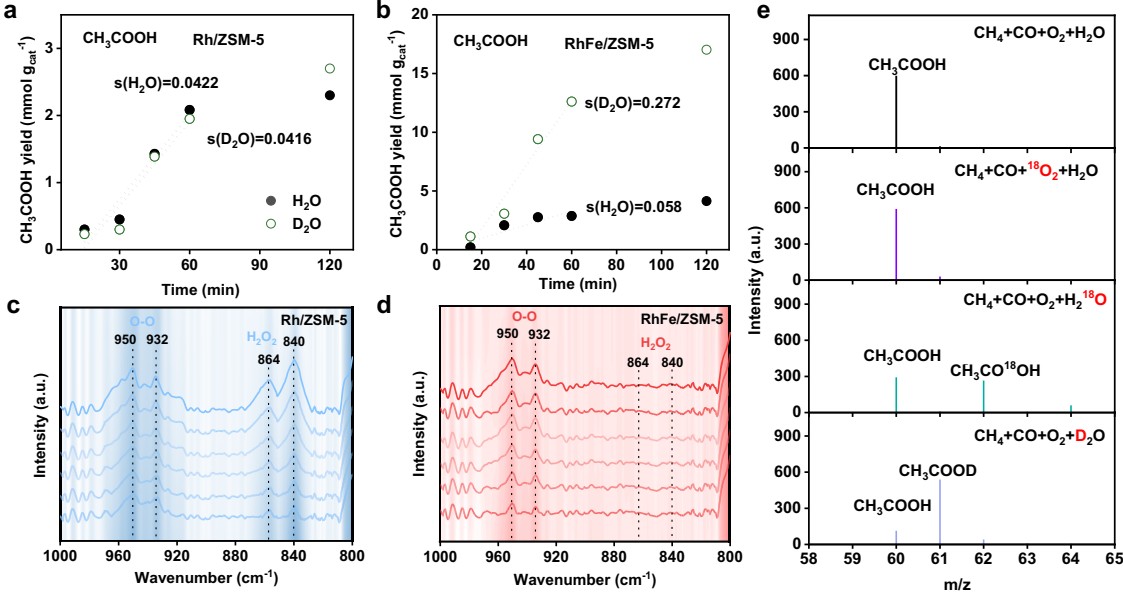

**Fig. 3 | CH$_4$ adsorption and activation mechanism. a** CH$_3$COOH production under H$_2$O and D$_2$O with different reaction time over Rh/ZSM-5. **b** CH$_3$COOH production under H$_2$O and D$_2$O with different reaction time over RhFe/ZSM-5. **c, d** Detection of H$_2$O$_2$ using in-situ DRIFTS spectra of Rh/ZSM-5 and RhFe/ZSM-5 catalysts in the presence of O$_2$, CO and H$_2$O at 463 K. **e** GC-MS spectra of the isotope CH$_3$COOH produced from CH$_4$ conversion when using CH$_4$ + CO + $^{16}$O$_2$ + H$_2$$^{18}$O, CH$_4$ + CO + $^{18}$O$_2$ + H$_2$$^{16}$O and CH$_4$ + CO + O$_2$ + D$_2$O as the reactants.

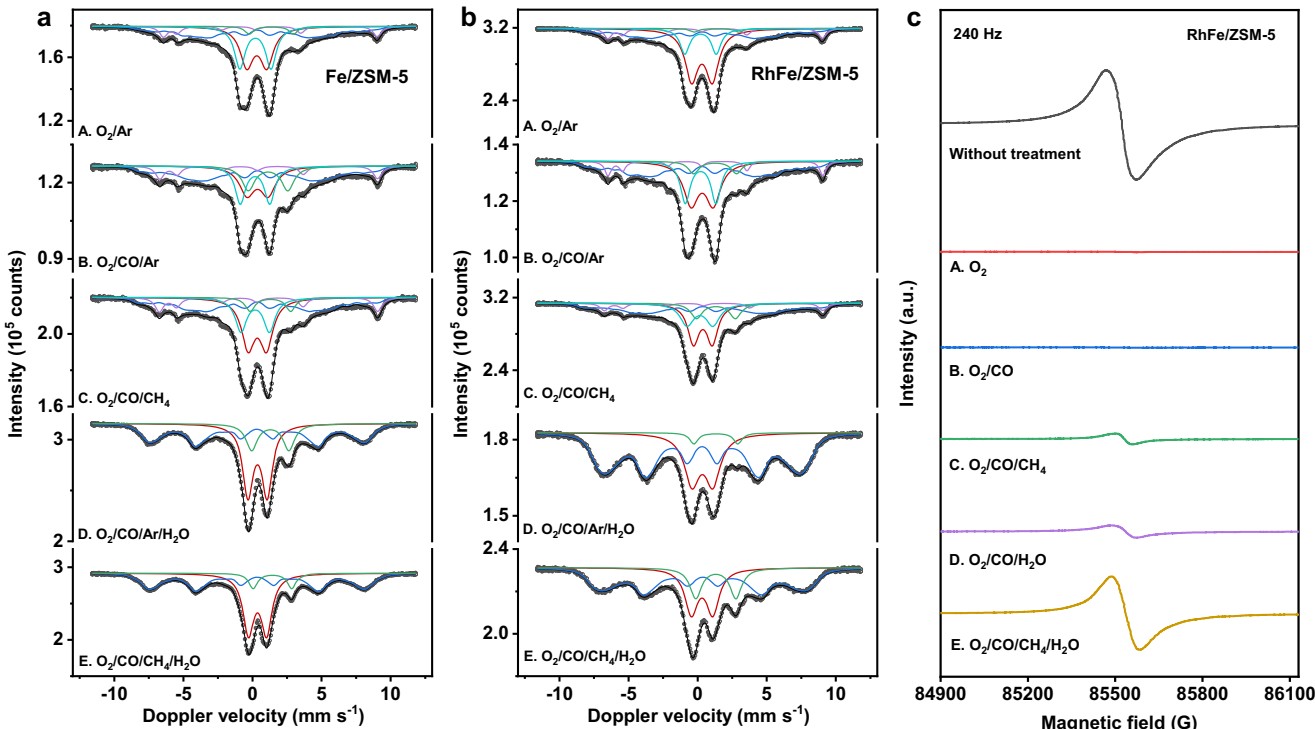

**Fig. 4 | In-situ Mössbauer spectroscopy. a** Fe/ZSM-5. **b** RhFe/ZSM-5. Transmission $^{57}$Fe Mössbauer spectra were collected at 4.2 K with a sinusoidal velocity spectrometer using a $^{57}$Co(Rh) source. Velocity calibration was carried out using an α-Fe foil at room temperature. The source and the absorbing samples were kept at the same temperature during the measurements. **c** In-situ High-field EPR of the catalyst during reaction. Reaction conditions: 463 K, 12 h, test conditions: 15 K, microwave frequency: 240 Hz.

Fe/ZSM-5 and RhFe/ZSM-5 (Supplementary Table 6, 7), having low isomer shift values, in the 0.14-0.21 mm/s range ($Fe^{(IV)}=O$ (I) and $Fe^{(IV)}=O$ (II), represent $O=Fe^{(IV)}-O-Fe^{(IV)}=O$, and highly dispersed $Fe^{(IV)}=O$, respectively, indicating $O_2$ facilitates the $Fe^{(IV)}=O$ formation[32–36]. Additionally, the presence of a doublet signal at 4.2 K indicates the spin relaxation effects in highly dispersed species-the doublet (cyan) with 0.21 mm/s isomer shift being assigned to diiron (IV) complexes, $O=Fe^{(IV)}-O-Fe^{(IV)}=O$, in the Fe/ZSM-5[37]. Simultaneously, the observation of a well-defined sextuplet signal at 4.2 K (Fig. 4a) (magenta sub-spectrum) with such highly dispersed species indicates the presence of superexchange interactions over long-range, mediated by the oxygen atoms[38]. The superexchange interactions change the spin state at the oxoiron (IV) sites[39], as evidenced by the lower measured isomer shift values (0.14–0.16 mm/s), suggesting the presence of high-spin $Fe^{(IV)}$ species. This result is consistent with the high-field EPR (HF-EPR) spectra (15 K, 240 Hz) (Fig. 4c Treatment A) after 12 h $O_2$ treatment at 463 K, where the disappearance of Fe and Rh signals coincided with the oxidation of Fe to $Fe^{(IV)}$ (S = 2) and Rh to $Rh^{(III)}$ (S = 0) (Supplementary Fig. 36).

The second dimeric doublet (Fig. 4a, red colour) in the Mössbauer spectra measured after the $O_2$/Ar treatment has an isomer shift value (0.31 mm/s), which is smaller than that measured in the fresh samples (0.53/0.50 mm/s minus 0.14 mm/s correction). This could indicate the formation of mixed dimeric complexes like $HO-Fe^{(III)}-O-Fe^{(IV)}=O$ in Fe/ZSM-5[39]. This component is higher in the Rh-containing sample (43 vs. 30 %), due to the partial formation of mixed Rh-O-Fe species. Also, the quadrupole splitting values are relatively different in these dimeric doublet signals (1.44 vs. 1.50 mm/s), indicating higher charge asymmetry around the Fe atoms in the Rh-containing zeolite. Under the $O_2$/Ar reaction conditions, the content of $Fe^{(III)}$-D in RhFe-ZSM-5 (43%) is higher than that in Fe-ZSM-5 (30%). This is due to Rh's ability to donate electrons to Fe. XPS and EXAFS analysis confirm this electron transfer process, further indicating that $HO-Fe^{(III)}-O-Rh$ species constitute a significant proportion of $Fe^{(III)}$-D in Fe(Rh)-ZSM-5.

Upon CO introduction (Fig. 4a, b, Treatment B), the signal of dimeric iron $Fe^{(III)}$-D decreased in the Mössbauer spectra, while the signal of dispersed iron $Fe^{(III)}$-PHS (blue) increased. This suggests that CO preferentially reacts with the bridging oxygen in $HO-Fe^{(III)}-O-Rh(Fe)$ dimers rather than $Fe^{(IV)}=O$ (I) and $Fe^{(III)}$-D. The selective reaction of CO with the bridging oxygen disrupts long-range superexchange interactions mediated by oxygen atoms, leading to the conversion of $Fe^{(III)}$-D into $Fe^{(III)}$-PHS. The amount of $Fe^{(IV)}$ showed little change, suggesting its low reactivity in oxidizing CO, which aligns well with the EPR results (Fig. 4c, treatment B).

When $CH_4$ was introduced (Fig. 4a, b, Treatment C), the spectral contributions of $Fe^{(IV)}$ species remained largely unchanged of RhFe/ZSM-5 in Mössbauer spectroscopy, but their hyperfine parameters of $Fe^{(IV)}$ shifted more significantly than that Fe/ZSM-5, suggesting that the neighboring Rh sites play a crucial role in C-H activation, rather than the $Fe^{(IV)}=O$ species. Upon $CH_4$ introduction to the RhFe/ZSM system post $O_2$/CO treatment, the HF-EPR signal re-emerged (Fig. 4c Treatment C), demonstrating that $CH_4$ can be activated by high-valent metals ($Fe^{(IV)}$ or $Rh^{(III)}$). Importantly, the Mössbauer spectra revealed no significant change in $Fe^{(IV)}$ species after $CH_4$ exposure, confirming that the observed signal in the HF-EPR originates from the reduction of $Rh^{(III)}$ to $Rh^{(II)}$, while $Rh^{(III)}$ actively participated in $CH_4$ activation.

Subsequent $H_2O$ treatments (Fig. 4a, b, Treatment D) appear to reset the samples to their fresh state. However, the dimeric complex in the RhFe/ZSM-5 remained with low isomer shift values (0.32-0.34 mm/s), indicating the continued presence of mixed $Fe^{(III)}$-O-Rh species. The $Rh-O_2-H_2O$ system (Supplementary Fig. 37) also exhibited a $Rh^{(II)}$ signal, proving that both $Fe^{(IV)}$ and $Rh^{(III)}$ can activate water. Notably, the signal amplitude increased markedly upon $H_2O$ addition, indicating that high-valent metals extract •H radicals (not protons) from $H_2O$, concurrently generating •OH radicals.

Furthermore, in-situ infrared spectroscopy was employed to capture the key intermediates in $CH_4$ activation and coupling

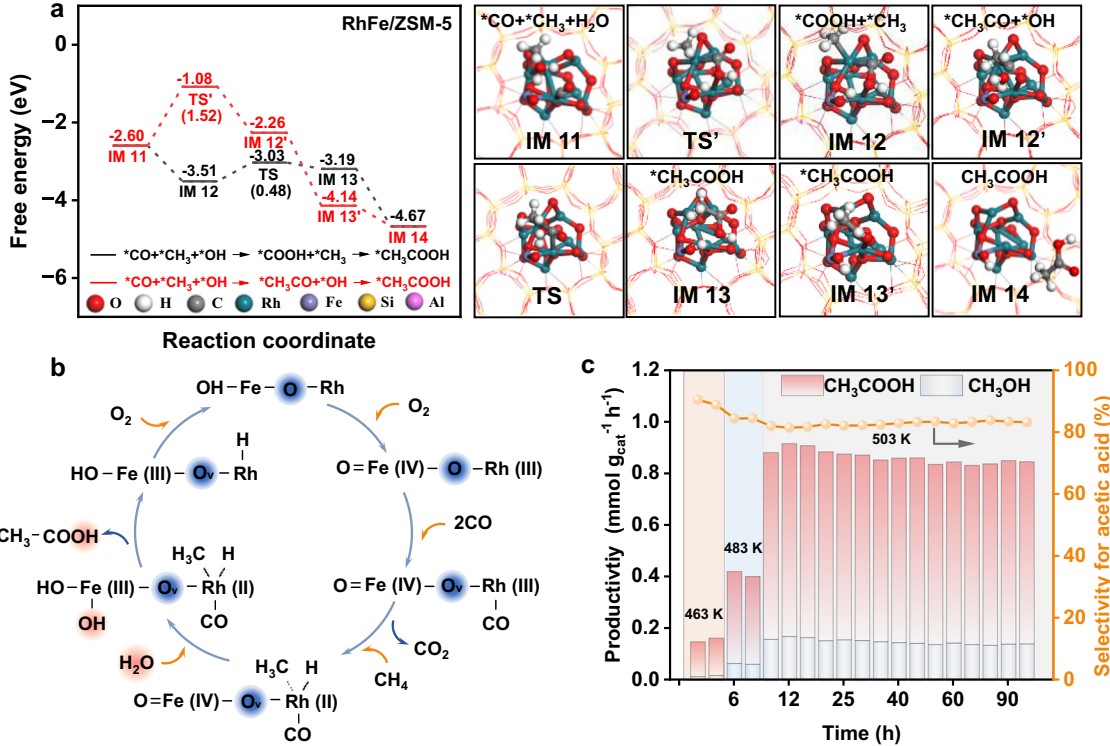

**Fig. 5 | DFT calculations and stability in a continuous flow reactor. a** DFT calculations of *COOH and *CH₃CO path. **b** Proposed RhFe/ZSM-5 catalytic cycle. **c** The CH₄ oxidative coupling reaction was catalyzed in a continuous regime. Reaction conditions: Before the reaction, 0.2 g of RhFe/ZSM-5 catalyst (20–40 mesh) was thoroughly mixed with 0.6 g of acid-washed quartz sand (20–40 mesh) to minimize temperature gradients, and the mixture was loaded into a stainless-steel fixed-bed reactor. 503 K, H₂O 0.3 mL/min, CH₄ 20 mL/min, CO 10 mL/min, O₂ 5 mL/min.

(Supplementary Fig. 38). Water vapor was introduced into the in-situ cell using an N₂ carrier gas, and a baseline was recorded after stabilization. Sequential addition of CH₄ and CO yielded no significant signals corresponding to products such as *C = O in CH₃COOH. However, upon O₂ introduction, the spectrum underwent significant changes, revealing signals for •CH₃ and *C = O at 1467 cm⁻¹ and 1743 cm⁻¹, respectively[40,41]. Notably, the water consumption peaks at 3670 cm⁻¹, coinciding with the results of Mössbauer and HF-EPR. EPR further detected the presence of •CH₃ radicals (Supplementary Fig. 39). In the in-situ FTIR spectra under condition of O₂/H₂O and O₂/H₂O /CO over Fe/ZSM-5 (Supplementary Fig. 40), the vibration of CO was observed in the range of 2011–2256 cm⁻¹, while the characteristic peaks at 1576 cm⁻¹ and 1352 cm⁻¹ correspond to the *COOH, indicating CO hydroxylation[42]. Compared with the O₂/H₂O/CH₄ system (Supplementary Fig. 41), O₂/CO/H₂O/CH₄ system shows significantly enhanced HF-EPR signals, demonstrating that CO promotes the conversion of H₂O and CH₄ into •OH and •CH₃ radicals, facilitating •COOH formation via •OH/CO coupling and subsequent CH₃COOH synthesis.

To further confirm the C-C coupling pathway, we conducted control experiments using methanol, formaldehyde, and formic acid as potential reactants under identical conditions. However, none of these compounds resulted in the formation of CH₃COOH (Supplementary Table 2, Entries 6-8), effectively ruling out a methanol-based pathway, such as the Monsanto process. When CH₃I was used as a substitute for CH₄, an increase in the yields of both CH₃COOH and CH₃OH was observed, indicating the involvement of •CH₃ radicals in the reaction (Supplementary Fig. 42).

Complementary DFT calculations further support a radical based pathway involving •CH₃ and •COOH coupling, which exhibits a significantly lower energy barrier than alternative routes. Specifically, two plausible coupling mechanisms were assessed: (i) *CH₃ coupling with *CO, followed by hydroxylation, and (ii) *CH₃ coupling directly with *COOH formed via *CO and H₂O activation. The former pathway faces a prohibitive barrier of 1.52 eV (Fig. 5a), whereas the latter proceeds with a much lower barrier of 0.48 eV, aligning well with the experimental observation of *COOH intermediates. The overall reaction mechanism is summarized in Fig. 5b, Supplementary Fig. 43, and Supplementary Table 8. Together, these results highlight *COOH mediated C-C coupling as the energetically and mechanistically preferred carbonylation route.

## Catalyst performance and stability
Through mechanistic insights into CH₄ oxidation, we demonstrate that supported RhFe/ZSM-5 catalysts exhibit exceptional stability in acetate acid production during batch reactions (Supplementary Fig. 44). This performance was successfully translated to continuous-flow operation using RhFe/ZSM-5 in CH₄ oxidative carbonylation (Supplementary Fig. 45). In the flow reactor at 503 K (Fig. 5c), the RhFe/ZSM-5 catalyst maintained stable activity for 100 hours, achieving a consistent CH₃COOH productivity of 0.71 mmol g₍cat₎⁻¹ h⁻¹ with 83% selectivity and no observable deactivation. Meanwhile, the Rh/ZSM-5 catalyst showed significantly lower performance, yielding only 0.14 mmol g₍cat₎⁻¹ h⁻¹ with 68% CH₃COOH selectivity (Supplementary Fig. 46). Comprehensive post-reaction of batch and continuous characterization revealed preserved crystallinity and valence state by XRD and XPS (Supplementary Figs. 47, 48), and STEM confirmed maintained dispersion of both noble metal nanoparticles and atomic Fe sites all structural features that directly account for the outstanding catalytic stability observed (Supplementary Figs. 49, 50).

## Discussion
In this work, we present a spatially engineered RhFe/ZSM-5 catalyst that enables efficient oxidative carbonylation of CH₄ to CH₃COOH via a distinctive synergy between radical generation and coupling. Unlike reported systems that rely on complete WGS cycles, this catalyst

leverages $O_2$ to convert Fe sites into $Fe^{(IV)} = O$ species, initiating a truncated WGS-like pathway. This process facilitates the direct activation of $H_2O$ to generate •OH. Simultaneously, $Rh^{(III)}$ centers selectively activate $CH_4$ to produce •$CH_3$.

A central innovation lies in the rapid and selective coupling of these radical intermediates within the confined micropores of the ZSM-5 framework. The •OH species react swiftly with CO to form •COOH intermediates, which in turn couple with Rh-derived •$CH_3$ radicals to yield $CH_3COOH$. This well-coordinated cascade circumvents the kinetic limitations of traditional CO insertion routes and enables a high $CH_3COOH$ selectivity of 92% at 463 K, with a productivity of 18.2 mmol $g_{cat}^{-1}$ $h^{-1}$, 5.7 times that of Rh-only systems. Furthermore, the catalyst exhibits excellent long-term stability over 100 hours of continuous operation, underscoring its practical viability.

Overall, this work establishes a new paradigm for low-temperature $CH_4$ valorization by integrating spatially resolved bimetallic active sites with an unconventional, WGS-inspired water activation pathway. The mechanistic insights and design principles provided here offer a promising foundation for the development of next-generation catalytic systems for the direct transformation of $CH_4$ into value-added $C_{2+}$ chemicals.

## Methods

### Materials and chemicals

Ferrous chloride ($FeCl_2$, AR, 99%, Macklin), Rhodium nitrate solution ($Rh(NO_3)_3$, AR, 5% in $H_2O$, Macklin), and other metal chlorides (Cu, Au, Pd) (AR, 99%, Macklin) were used as metal precursors. Silica solution ($SiO_2$, 29–31%, Macklin), Sodium hydroxide (NaOH, AR, Macklin), sodium aluminate ($NaAlO_2$, AR, Macklin). Methane, carbon monoxide, argon, and oxygen (99.999 vol.%, Qingdao Xin ke yuan) were used as the feedstock gases. All chemicals were used as received without any further purification. Deionized water was used throughout the research.

### Catalyst preparation

RhFe/ZSM-5 was synthesized using a previously reported seeded growth template-free method with slight modifications. This process included two steps: the synthesis of ZSM-5 seeds and the synthesis of RhFe/ZSM-5.

In the first step, 15 g of colloidal silica was dissolved in 7 mL of a NaOH solution (1 M) under stirring at 373 K for 1 h. A separate solution was prepared by dissolving 0.45 g of sodium aluminate in 7 mL of a NaOH solution (1 M). The two solutions were mixed to obtain a synthetic aluminosilicate gel with a molar composition of 4 $Na_2O$:1 $Al_2O_3$:36 $SiO_2$:460 $H_2O$. The gel was stirred vigorously at 373 K for 2 h and transferred to a stainless-steel autoclave for crystallization at 453 K for 48 h. The obtained crystals were collected by filtration, washed with deionized water, and dried at 373 K to produce the ZSM-5 seeds.

In the second step, a synthetic aluminosilicate gel was prepared using the same method as in the first step. Subsequently, 1.8 mL of $Rh(NO_3)_3$ solution (5% in $H_2O$), containing 0.12 g of ferrous chloride ($FeCl_2$) and 0.06 g of seeds, was added sequentially to the gel. After stirring for 30 min, the mixture was transferred to an autoclave for crystallization at 453 K for 48 h. The obtained crystals were collected by filtration, washed with deionized water, and dried at 353 K to obtain the as-synthesized RhFe/ZSM-5. Finally, RhFe/ZSM-5 was converted to its H-form via ion exchange with ammonium chloride (0.5 M) for 12 h, followed by calcination in air at 823 K for 6 h. The different metal loading in ZSM-5 could be tuned by varying the types of metal precursors while keeping the other conditions unchanged.

### Characterizations

X-ray diffraction (XRD) was performed by the diffractometer (X'Pert PRO MPD, PANalytical, Netherlands) with Cu Kα radiation (40 kV, 100 mA, $\lambda = 0.154$ nm). The scanning electron microscope (SEM, JSM-7500F, Japan) was utilized to observe material morphology. HRTEM

and EDS-mapping images were captured on the Tecni G30 instrument (FEI, USA). The morphology of the samples was further observed by aberration corrected high-angle annular dark field scanning transmission electron microscope (AC-HAADF-STEM, Themis Z, Thermo Scientific, USA). The iDPC-STEM experiments were performed at 300 kV on a FEI Themis Z microscope equipped with two aberration correctors. The content of metal elements was determined by the inductively coupled plasma atomic emission spectroscopy (ICP-AES, Agilent 730, USA). The information on the electronic state of the material surface was collected via the X-ray photoelectron spectrometer (XPS, ESCA-LAB 250Xi, Thermo Scientific, USA), and all binding energies were calibrated to the C1s peak of surface-contaminated carbon at 284.8 eV. Raman spectra were measured on the instrument (InVia Reflex, Renishaw, England). X-ray Absorption Spectroscopy (XAS) data for Fe and Rh K-edge were collected at the BL13SSW station of the Shanghai Synchrotron Radiation Facility (SSRF)[43], and data were obtained in fluorescence excitation mode using a Lytle detector. The X-ray Absorption Near Edge Structure (XANES) and Fourier-transformed Extended X-ray Absorption Fine Structure (EXAFS) data were analyzed using ATHENA and ARTEMIS software, respectively, and MATLAB software was employed for the analysis of wavelet transformed EXAFS data. UV-Vis diffuse reflectance spectra were obtained from the spectrometer (UV-2700, Shimadzu, Japan) furnished with an integrating sphere device. The fluorescent spectra were collected by the fluorescence spectrophotometer (RF-6000, Shimadzu, Japan). In-situ diffuse reflectance infrared Fourier transform spectroscopy (DRIFTS) measurements were measured on the instrument (VERTEX70, Bruker, Germany), the mercury cadmium telluride (MCT) detector was adopted, $H_2O$ was bubbled into the reaction by $CH_4$, $O_2$, and CO, when the temperature was raised and stabilized to 463 K. The background correction was stabilized, and $CH_4$, $O_2$, and CO replaced Ar for testing. Electron paramagnetic resonance (EPR) spectroscopy measurement was performed on the instrument (Bruker EMXplus, Germany) with 5,5-dimethyl-1-pyrroline-N-oxide (DMPO) as the radical trap. The samples were dispersed in $H_2O_2$ aqueous solution, dissolved $CH_4$ and CO to detect •$CH_3$ and •OH. The gas chromatography (Scion 456 C, Tianmei, China) is equipped with a thermal conductivity detector (TCD), two flame ionization detectors (FID), a methanizer, and a headspace autosampler (DK-5001A, Beijing Zhongxing, China), which were used to quantify gaseous and $CH_3OH$ products. High-performance liquid chromatography (HPLC, Prominence-i, LC-2030 Plus, Japan) equipped with a Refractive Index Detector (RID) was used to quantify HCOOH and $CH_3COOH$ products. Transmission $^{57}Fe$ Mössbauer spectra were collected at 4.2 K with a sinusoidal velocity spectrometer using a $^{57}Co$(Rh) source. Velocity calibration was carried out using an α-Fe foil at room temperature. The source and the absorbing samples were kept at the same temperature during the measurements. The Mössbauer spectra were fitted using the Mosswinn 4.0 program[44]. The experiments were performed in a state-of-the-art high-pressure Mössbauer in-situ cell - developed at Reactor Institute Delft[45].

### Oxidative coupling of methane

The methane carbonylation experiment was carried out in a 50 mL high-pressure reactor (Shi ji shen lang). The catalyst (10 mg) was uniformly dispersed in 20 mL of distilled water and sonicated for 15 min. The reactor was purged with argon gas to replace the air for 3–5 times. Then, methane, oxygen, and carbon monoxide were injected at the required pressure. The reaction was carried out at 463 K for 2 h. After the reaction, when the reactor temperature is reduced to room temperature and then cooled to below 283 K with an ice bath, the gas and liquid were collected.

### Cyclic experiment

The recycle test followed the same procedure. After each run, the spent catalyst was separated using filtration, washed with a large

amount of $H_2O$, and then dried at 353 K in the vacuum oven for the next cycle. The repeated experiments were carried out under the same experimental conditions to verify the stability of the catalyst.

### •OH detection experiment

Terephthalic acid was utilized as the probe for the detection of •OH via the production of 2-hydroxyterephthalic acid. Typically, 15 mg catalyst was dispersed in 20 mL terephthalic acid solution (0.5 mM). After 2 h, a certain amount of the reactant was measured on the spectro-fluorometer (RF-6000, Shimadzu, Japan) after filtering. The excitation wavelength was 315 nm, and the peak value at 425 nm was observed to semi-quantify the concentration of •OH radicals.

### The detection of residual $H_2O_2$

After the reaction, 0.1 mol/L $H_2SO_4$ and 0.05 mol/L potassium titanium oxalate were added to the solution, stirred for one minute, and the absorbance was measured by UV.

### Mössbauer spectroscopy in-situ study

Transmission $^{57}Fe$ Mössbauer spectra were collected at 4.2 K with a sinusoidal velocity spectrometer using a $^{57}Co(Rh)$ source. Velocity calibration was carried out using an α-Fe foil at room temperature. The source and the absorbing samples were kept at the same temperature during the measurements. The Mössbauer spectra were fitted using the Mosswinn 4.0 program. The experiments were performed in a state-of-the-art high-pressure Mössbauer in-situ cell developed at Reactor Institute Delft[45].

### DFT

The spin polarization DFT calculations were conducted using the QUICKSTEP module of the CP2K-2024.3 code[46], which utilizes a combination of Gaussian and plane-wave basis sets. For the exchange-correlation functional, the Perdew-Burke-Ernzerhof (PBE) approach was implemented. Core electrons were modeled with Goedecker-Teter-Hutter pseudopotentials, while the wave functions of valence electrons were represented using a DZVP-MOLOPT-SR-GTH basis set. 500 Ry cutoff and 60 Ry rel_cutoff were applied. During the calculations, all the atomic positions were fully relaxed until the force was smaller than $4.5 \times 10^{-4}$. The structures were optimized using the Broyden-Fletcher-Goldfarb-Shanno (BFGS) algorithm, setting the SCF convergence criterion to $1.0 \times 10^{-6}$. To account for van der Waals (vdW) interactions, the DFT-D3(BJ) method, which includes an empirical damping term, was added to the electronic energy calculations.

The free energy G is calculated by $G = E + ZPE - TS$, where E is the DFT-based energy, ZPE and TS are the correction of zero point energy and entropy, respectively. Input files and data post-processing were performed using the Multiwfn program[47,48].

## Data availability

Source data are provided with this paper.

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

## Acknowledgements

This work was supported by National Natural Science Foundation of China (22322815, 22561142234, 22179146, and 22138013) (W.W), Shandong Provincial Natural Science Foundation (Grant No. ZR2025QA14), the Fundamental Research Funds for Central Universities (18CX07009A), Independent Innovation Research Project (Science and Engineering) (20CX06072A). We thank the BL13SSW beamline (31124.02.SSRF.BL13SSW) at the Shanghai Synchrotron Radiation Facility for the XAFS.

## Author contributions

W.W. conceived and supervised the project. G.J.H. and M.W. co-supervised the project. H.Z., R.J.L., and A.I.D. conducted most experiments, including synthesis, characterization, and testing, as well as data analysis. Y.L. (Yang Li) carried out the DFT calculations section. S.W. performed characterization and testing analysis. Z.W. conducted spectrum analysis. J.Z. contributed to the data analysis of the XAS spectra. N.F.D supervised the project. Y.X. conducted DRIFTS analysis. Y.L. (Yunyun Li) conducted an analysis of the mass spectrometry. T.E.D. provided advice and expertise. H.Z., W.W., and G.J.H. wrote and revised the paper. All authors discussed the paper.

## Competing interests

The authors declare no competing interests.
