## [Transparent Peer Review file · Nature Communications]

Direct oxidative carbonylation of methane to acetic acid via high-valent iron-oxo mediated water activation

Corresponding Author: Professor Graham Hutchings

Version 0:

Reviewer comments:

Reviewer #1

(Remarks to the Author)

The manuscript "Direct oxidative carbonylation of methane to acetic acid via high-valent iron oxo mediated water oxidation" reports on the use of Rh–O–Fe sites in ZSM-5 for the oxidative carbonylation of methane to acetic acid. The authors claim to introduce a novel paradigm for selective methane carbonylation, achieving an acetic acid production rate of 18.2 mmol g_{cat}⁻¹ h⁻¹ with 92% selectivity. However, there appears to be overlap with recent studies on Fe- and Rh-modified zeolites for methane carbonylation (doi.org/10.1002/cctc.202000168, <https://doi.org/10.1016/j.cattod.2025.115558>, <https://doi.org/10.1016/j.micromeso.2021.111581>, <https://doi.org/10.1134/S0965544123060075>). These papers achieved comparable production rates and selectivities through radical-based coupling mechanisms. Thus, although the present study demonstrates improved catalytic performance, it does not appear to represent an entirely new advance in the field. Additional points that require clarification or improvement:

1. The reaction should inevitably be accompanied by CO oxidation to CO₂; however, the authors do not report selectivity toward gas-phase products.
2. Stability tests are presented up to 100 h but only in continuous flow at much lower productivity than in batch. Why is there such a large discrepancy between batch (18.2 mmol g_{cat}⁻¹ h⁻¹) and flow (0.71 mmol g_{cat}⁻¹ h⁻¹)? This requires explanation.
3. Results are reported per gram of catalyst, but it would be also important to compare catalytic activity based on TOF values per metal site.
4. The mechanistic assignment relies heavily on Mössbauer and EPR spectroscopy, but the interpretation of Fe(IV)=O as responsible for the direct conversion of H₂O to OH radicals seems speculative. The role of the zeolite framework and its acid sites is also unclear.
5. The EXAFS assignments of peaks to Fe–O–Fe and Fe–O–Rh are not fully convincing. For such assignments, reference samples with the proposed bonding environments are essential.
6. The respective roles of Rh and Fe remain ambiguous: the text suggests Rh(III) activates CH₄ while Fe(IV)=O activates H₂O, yet Mössbauer data indicate Fe species are not strongly affected by CH₄. More clarification is needed.
7. No direct detection of CH₃ or COOH radicals intermediates is shown by EPR. Radical trapping experiments are necessary to confirm the presence of these species.

Reviewer #2

(Remarks to the Author)

Typical metals like Rh and Fe have been widely reported for methane oxidative carbonylation to produce acetic acid in both CH₄/CO/O₂ and CH₄/CO/H₂O₂ systems. Rh-Fe dual-atom sites on MoS₂ and Ir–O–Ir dimeric sites on SBA-15 have achieved high acetic acid selectivity at low temperatures (J. Am. Chem. Soc. 2025, 147, 14530; J. Am. Chem. Soc. 2023, 145, 769). Here, the authors report Rh–O–Fe dual-metal sites that enable CH₃COOH synthesis at higher temperatures (463 K) with high yield and selectivity, typically shows low catalytic activity below 423 K. Unlike the conventional CO insertion into M–CH₃, they propose a direct carbon-carbon coupling mechanism, offering a more feasible pathway for acetic acid formation, similar pathway has also been reported (Appl. Catal. B 2025, 378, 125632). Although the findings sound good and the manuscript is meticulously prepared, the overall novelty of this study appears constrained by prior publications in this field. Additionally, several substantive concerns preclude my endorsement for publication in Nature Communications in its current form.

1. The assertion of Rh-O-Fe structure formation relies primarily on indirect evidence from Fe/Rh K-edge FT-EXAFS and ^{57}Fe Mössbauer spectroscopic analyses. However, these techniques cannot provide unambiguous structural confirmation. To strengthen their claim, the authors should incorporate direct visualization through aberration-corrected HAADF-STEM and additional spectroscopic validation via CO-IR measurements.
2. The authors propose a reaction mechanism for CH_3COOH formation involving direct carbon-carbon coupling between $\cdot\text{CH}_3$ and $\cdot\text{COOH}$ radicals, same as previous study (Angew. Chem. Int. Ed. 2024, 63, e202315343; Appl. Catal. B 2025, 378, 125632). However, this claim requires more substantial evidence. First, the manuscript fails to provide direct experimental evidence for $\cdot\text{CH}_3$ radical capture (EPR). Second, while $\cdot\text{COOH}$ intermediates are known to form in CO/O₂/H₂O systems (ACS Catal. 2020, 10, 13993). From Fig. 3C, the formation of H₂O₂ is clearly observed, indicating that Rh sites can indeed catalyze the reaction of CO, O₂, and H₂O to produce H₂O₂ via $\cdot\text{COOH}$ intermediates. However, Fig. 5B presents a strikingly distinct O₂ activation pathway on Rh sites for methane oxidation. The authors should provide more evidence about this phenomenon.
3. The coupling of $\cdot\text{COOH}$ with $\cdot\text{CH}_3$ species to produce acetic acid needs verification. As a proof of concept, if CH₃I (provided CH₃ radicals) were added as substrate, significantly enhanced acetic acid production would be expected. Furthermore, following the proposed mechanism, using ethane as substrate should yield substantial amounts of $\text{CH}_3\text{CH}_2\text{COOH}$.
4. The authors propose that the in situ formation of high-valent Fe(IV)=O initiates a truncated WGS pathway, wherein H₂O is directly dissociated into $\cdot\text{OH}$. These radicals rapidly react with CO to form $\cdot\text{COOH}$ intermediates, which then couple with $\cdot\text{CH}_3$ to yield CH_3COOH . To validate this mechanism, in situ DRIFTS studies on Fe/ZSM-5 are recommended: Upon sequential exposure to O₂ and CO, the formation of $\cdot\text{COOH}$ intermediates should be observable.

Reviewer #3

(Remarks to the Author)

In this article, Zhang and coauthors presented Rh-Fe site-supported ZSM-5 towards selective CH₄ conversion to CH₃COOH. The well-designed RhFe/ZSM-5 exhibits quite high CH₃COOH production rate 18.2 mmol gcat⁻¹ h⁻¹, with 92% selectivity, which is much competitive among the current promising catalysts. Great efforts have been devoted to the study on the characterization and involved catalytic conversion process by in-situ Mössbauer spectroscopy, in-situ high-field EPR and in-situ FT-IR, etc. In particular, the mechanism of the $\cdot\text{COOH}$ and $\cdot\text{CH}_3$ to yield CH₃COOH under Rh(III)/Fe(IV)=O active site was revealed. Overall, the conclusion is supported by solid evidence from both comprehensive experimental results and theoretical calculation, with adequate novelty for the material designing, synthesis and insights validation. Hence, I would like to recommend this work to be published after minor revision.

1. This paper presents a sustainable methane conversion strategy. By comparing it with existing approaches, the work highlights the advantages of the proposed catalytic method, thereby strengthening the manuscript's overall contribution.
2. The notation for Fe(IV)/Fe⁴⁺ and Rh(III)/Rh³⁺ should be consistent throughout the text.
3. Through extensive comparative experiments, the authors provide a detailed demonstration of the RhFe/ZSM-5 catalyst's effectiveness. Beyond the comparison of different metals, have the authors considered evaluating other molecular sieves, such as MOR, SSZ-13, or SiO₂?
4. The authors evaluated the catalyst's performance under pressurized conditions. However, further studies under ambient conditions would be beneficial, as they facilitate easier scaling and mitigate the safety concerns of handling high-pressure methane-oxygen mixtures.
5. The authors convincingly demonstrated the existence of an Rh-O-Fe structure in the catalyst through EXAFS and ^{57}Fe Mössbauer spectra. Have they considered how the catalytic performance would be affected if the Rh-O-Fe structure were absent, or if Rh and Fe were spatially separated? Such an investigation could further corroborate the crucial role of the Rh-O-Fe structure in the oxidative coupling of methane, which would be a very interesting demonstration.

Version 1:

Reviewer comments:

Reviewer #1

(Remarks to the Author)

The authors have addressed almost all of my comments. However, it is clear that, beyond novelty, a significant issue is the substantial formation of CO₂ during the reaction of CO oxidation, reaching up to 50 mmol g⁻¹ h⁻¹, which is even higher than the amount of acetic acid produced. This point must be addressed in the manuscript before acceptance. The amount of CO₂ formed is almost twice that of acetic acid, and the text should therefore be corrected, in particular the statement: "In addition, the similar amount of CO₂ generated...". The authors should report the selectivity to acetic acid either by explicitly accounting for CO₂ formation or by clearly distinguishing between liquid-phase selectivity and overall selectivity including CO₂.

Reviewer #2

(Remarks to the Author)

Further revision has been made to improve the manuscript and it is suggested for acceptance.

Reviewer #3

(Remarks to the Author)

The authors provided detailed responses to the questions raised by the reviewers. The quality of the manuscript has significantly improved. Therefore, I recommend publishing this manuscript in Nature Communications.

Dear Reviewers,

Many thanks for the time and effort you have dedicated to reviewing our manuscript. Based on your valuable comments and suggestions, we have carefully revised the manuscript, supplementing relevant characterization data and discussion. The changes made are highlighted in the revised manuscript and SI in yellow. We greatly appreciate your feedback, which has significantly enhanced the quality of this paper. A point-by-point description of the revisions and our responses to the reviewers' comments is provided below.

Response to reviewers' comments:

Reviewer #1 (Remarks to the Author):

The manuscript “Direct oxidative carbonylation of methane to acetic acid via high-valent iron oxo mediated water oxidation” reports on the use of Rh-O-Fe sites in ZSM-5 for the oxidative carbonylation of methane to acetic acid. The authors claim to introduce a novel paradigm for selective methane carbonylation, achieving an acetic acid production rate of $18.2 \text{ mmol g}_{\text{cat}}^{-1} \text{ h}^{-1}$ with 92% selectivity. However, there appears to be overlap with recent studies on Fe and Rh modified zeolites for methane carbonylation,

<https://doi.org/10.1002/cctc.202000168>,

<https://doi.org/10.1016/j.cattod.2025.115558>,

<https://doi.org/10.1016/j.micromeso.2021.111581>,

<https://doi.org/10.1134/S0965544123060075>.

These papers achieved comparable production rates and selectivities through radical-based coupling mechanisms. Thus, although the present study demonstrates improved catalytic performance, it does not appear to represent an entirely new advance in the field. Additional points that require clarification or improvement:

1. The reaction should inevitably be accompanied by CO oxidation to CO₂; however, the authors do not report selectivity toward gas-phase products.

Response:

We thank the reviewer for raising this important point regarding the selectivity toward gas-phase products, particularly the potential formation of CO₂ via CO

oxidation. We fully agree that a comprehensive analysis of gas phase products is essential for evaluating the overall selectivity and atom economy of the reaction. In response to the reviewer's comment, we have conducted additional gas phase product analysis under the different reaction conditions, using gas chromatography equipped with both TCD and FID detectors. The results are summarized below and included in the revised manuscript (Fig. S12-13).

Under optimized reaction conditions (463 K, 10 mg catalyst, 2 h reaction time, 20 mL H₂O, 30 bar CH₄, 3 bar O₂, 6 bar CO), the CO₂ production rate reached 40.6 mmol·g_{cat}⁻¹·h⁻¹. Gas-phase product analysis indicated that only CO₂ was detected in the system, with no other carbon-containing gaseous products observed. Similar amount of CO₂ generated, regardless of CH₄ presence, suggest that CO₂ is primarily from the oxidation of CO to CO₂ under these conditions.

Fig. S12. (a-e) The content of CO₂ under different catalysts, reaction conditions, 10 mg catalyst, 3 h, 20 mL H₂O, 30 bar CH₄, 3 bar O₂, 6 bar CO. (f) The performance of CO₂ under different temperature.

Fig. S13. Catalytic performance of CO₂. (A) Different H₂O volume; (B) Different amount of RhFe/ZSM-5 catalyst; (C) Different reaction time; (D) Different O₂ pressure; (E) Different CO pressure; (F) Different CH₄ pressure.

2. Stability tests are presented up to 100 h but only in continuous flow at much lower productivity than in batch. Why is there such a large discrepancy between batch (18.2 mmol g_{cat}⁻¹ h⁻¹) and flow (0.71 mmol g_{cat}⁻¹ h⁻¹)? This requires explanation.

Response: We thank the reviewer for raising this important point regarding the apparent discrepancy in acetic acid productivity between the batch and continuous flow reactors. The significant difference in the reported values primarily stems from fundamental distinctions in the reactor configurations and operating conditions, rather than indicating any inherent instability or performance loss of the catalyst in the flow system. We provide a detailed explanation below:

Reactor configuration:

The batch reactor operation was conducted in a liquid-phase slurry system with high-pressure gases (CH₄, CO, O₂) dissolved in water. This configuration often allows for high local concentrations of reactants near the catalyst surface within the confined reactor volume, potentially leading to higher observed initial reaction rates per mass of catalyst when measured over short periods (2 hours in our case).

The continuous flow stability test was performed in a gas-solid fixed-bed reactor, where the reactants (CH₄, CO, O₂, H₂O vapor) flow continuously over a stationary catalyst bed. This mode is more representative of potential industrial application and is essential for assessing long-term stability. However, the effective concentration of reactants contacting the catalyst at any given moment is typically lower than in a pressurized liquid-phase batch system.

Operating Conditions and Space Velocity:

The batch reactor productivity of 18.2 mmol g_{cat}⁻¹ h⁻¹ was achieved under optimized, intensive conditions (30 bar CH₄, 6 bar CO, 3 bar O₂, 463 K, 10 mg catalyst in 20 mL H₂O) designed to maximize the initial rate.

In the continuous flow reactor, the conditions were necessarily adjusted to ensure stable long-term operation. The weight of catalyst used in the flow reactor (0.2 g) was significantly larger than in the batch test (10 mg), and it was exposed to a continuous flow of reactants. The productivity in a flow reactor is inherently tied to the residence time and the available active sites per unit time. The lower reported value (0.71 mmol g_{cat}⁻¹ h⁻¹) reflects the steady-state productivity under these stable flow conditions. It is not directly comparable to the initial, high-intensity rate measured in the batch autoclave.

Purpose of the stability test:

The primary objective of the 100-hour continuous test was to demonstrate the robust stability and absence of deactivation of the RhFe/ZSM-5 catalyst under practical, steady-state conditions. The key finding is the stable activity maintained over 100 hours (Fig. 5C), which is a crucial metric for catalyst viability. The fact that the catalyst did not deactivate confirms its structural integrity, as corroborated by post-reaction characterization.

Inherent differences in productivity calculation:

Batch reactor productivity calculations can sometimes reflect initial, potentially mass-transfer-limited rates in a closed system. Continuous flow productivity represents a time-averaged, steady-state conversion in an open system, which is often lower on a per-gram-catalyst-per-hour basis but is a more reliable indicator for scalable processes.

In summary, the higher productivity in the batch reactor reflects an optimized, initial performance under specific intensive conditions in a liquid medium. In contrast, the lower but perfectly stable productivity in the continuous flow reactor demonstrates the catalyst's ability to function without deactivation under realistic, steady-state gas-solid reaction conditions over an extended period. This discrepancy is common in catalysis research when transitioning from batch screening to continuous flow operation and does not detract from the key finding of excellent catalyst stability.

3. Results are reported per gram of catalyst, but it would be also important to compare catalytic activity based on TOF values per metal site.

Response:

We thank the reviewer for this valuable suggestion. We agree that assessing the turnover frequency (TOF) provides a more fundamental metric for comparing the intrinsic activity of the catalytic sites. In response, we have calculated the TOF values based on the total moles of metal loaded (Rh, or Rh+Fe) (Fig. S5). This serves as a conservative and widely used estimation method, enabling a more direct comparison of site efficiency.

The calculation was performed using the acetic acid production rate under standard batch conditions (463 K, 10 mg catalyst, 2 h, 20 mL H₂O, 30 bar CH₄, 3 bar O₂, 6 bar CO) and the total metal content determined by ICP-AES (Table S3).

For the RhFe/ZSM-5 catalyst (0.55 wt% Rh, 0.25 wt% Fe), the TOF was calculated based on the sum of total Rh and Fe atoms. The resulting TOF value is ~216 h⁻¹. For the monometallic Rh/ZSM-5 catalyst (0.58 wt% Rh), the TOF calculated based on total Rh atoms is ~92 h⁻¹.

The TOF of the RhFe/ZSM-5 catalyst is approximately 2.5 times that of the Rh/ZSM-5 catalyst. This calculation, based on the total metal loading, provides clear evidence that the intrinsic activity per metal atom is significantly enhanced in the bimetallic system.

We have revised in the manuscript: For the RhFe/ZSM-5 catalyst (0.55 wt% Rh, 0.25 wt% Fe), the TOF calculated based on total Rh+Fe atoms reached ~216 h⁻¹, which

is ~2.5 times that of monometallic Rh/ZSM-5 (~92 h⁻¹, calculated on total Rh). This demonstrates a clear enhancement in intrinsic activity per metal atom in the bimetallic system (Fig. S5).

Fig. S5. TOF in Rh/ZSM-5 and RhFe/ZSM-5.

4. The mechanistic assignment relies heavily on Mössbauer and EPR spectroscopy, but the interpretation of Fe^(IV)=O as responsible for the direct conversion of H₂O to OH radicals seems speculative. The role of the zeolite framework and its acid sites is also unclear.

Response:

1. Clarification on the Role of Fe^(IV)=O in H₂O Activation

Our assignment of Fe^(IV)=O as the active species for H₂O cleavage is based on a combination of in-situ Mössbauer spectroscopy, isotopic labeling, and kinetic studies, which collectively provide a consistent picture of the reaction pathway.

In-situ Mössbauer Spectroscopy (Fig. 4):

After O₂ treatment, we observed the formation of Fe^(IV)=O species (isomer shift ~0.14-0.21 mm/s), which are stable under reaction conditions. Upon introduction of H₂O, the spectral features change in a manner consistent with the consumption of bridging oxygen species and the generation of OH-containing intermediates (e.g., HO-Fe^(III)). This transformation is more pronounced in the RhFe/ZSM-5 sample, indicating that Fe^(IV)=O is involved in H₂O dissociation.

Fig. 4. *In-situ* Mössbauer spectroscopy. (A) Fe/ZSM-5. (B) RhFe/ZSM-5. Transmission ^{57}Fe Mössbauer spectra were collected at 4.2 K with a sinusoidal velocity spectrometer using a $^{57}\text{Co}(\text{Rh})$ source. Velocity calibration was carried out using an $\alpha\text{-Fe}$ foil at room temperature. The source and the absorbing samples were kept at the same temperature during the measurements. (C) *In-situ* High-field EPR of the catalyst during reaction. Reaction conditions: 463 K, 12 h, test conditions: 15 K, microwave frequency: 240 Hz.

Isotopic Labeling (Fig. 3E):

When using $\text{H}_2^{18}\text{O} + ^{16}\text{O}_2$, the produced $\text{CH}_3\text{CO}^{18}\text{OH}$ contains ^{18}O in the hydroxyl group, confirming that the oxygen in the hydroxyl group originates from water, not O_2 . This rule out pathways involving O_2 derived peroxides as the primary oxygen source for $\bullet\text{OH}$ formation.

Fig. 3. (E) GC-MS spectra of the isotope CH_3COOH produced from CH_4 conversion when using $\text{CH}_4+\text{CO}+^{16}\text{O}_2+\text{H}_2^{18}\text{O}$, $\text{CH}_4+\text{CO}+^{18}\text{O}_2+\text{H}_2^{16}\text{O}$ and $\text{CH}_4+\text{CO}+\text{O}_2+\text{D}_2\text{O}$ as the reactants.

Kinetic Isotope Effect (KIE) Studies (Fig. 3B):

The inverse KIE ($K_{\text{H}}/K_{\text{D}} \approx 0.21$) observed for RhFe/ZSM-5 indicates that H_2O activation is not rate-determining step in the presence of Fe, consistent with a facile $\text{Fe}^{(\text{IV})}=\text{O}$ mediated H_2O cleavage step.

Fig. 3. (B) CH_3COOH production under H_2O and D_2O with different reaction time over RhFe/ZSM-5.

Control Experiments with H₂O₂ (Fig. S33-S34):

Adding H₂O₂ as an alternative •OH source resulted in lower acetic acid yield and selectivity, indicating that the Fe^(IV)=O pathway is more efficient and selective than free •OH generated from H₂O₂ decomposition.

Therefore, the assignment of the Fe^(IV)=O role is not speculative but represents the most consistent interpretation of the combined spectroscopic, kinetic, and isotopic data. We have further strengthened this discussion in the revised manuscript.

Fig. S33. The comparison of H₂O₂ with CO and O₂. Reaction conditions: 463 K, 10 mg catalyst, 2 h, 20 mL H₂O, 30 bar CH₄, 3 bar O₂/3 bar Ar as balance gas, 6 bar CO.

Fig. S34. The PL spectra of 2-hydroxyterephthalic acid using terephthalic acid as the probe molecule under O₂ + CO and H₂O₂.

2. The role of the zeolite framework and its acid sites.

We fully agree with the reviewer's observation that the role of acidic sites in the ZSM-5 framework was not sufficiently addressed in the original manuscript. To systematically elucidate their mechanism, we have supplemented the study with the following control experiments and characterizations, which further reveal the critical role of acidic sites in this reaction.

First, the functions of the ZSM-5 support are multifaceted, primarily manifested in the following two aspects:

1. Confinement effect and stabilization of active sites

The microporous structure of ZSM-5 provides spatial confinement, effectively stabilizing the RhFe active sites and inhibiting their aggregation. This has been confirmed by the excellent long-term stability of the catalyst and post-reaction characterization results.

(a). **Fig. S44** Cycle number of ReFe/ZSM-5 with five reactions. Reaction conditions: 463 K, 10 mg catalyst, 2 h, 20 mL H₂O, 40 bar CH₄, 3 bar O₂, 6 bar CO. (b) **Fig. 5C**. The CH₄ oxidative coupling reaction was catalyzed in a continuous regime. Reaction conditions: Before the reaction, 0.2 g of RhFe/ZSM-5 catalyst (20-40 mesh) was thoroughly mixed with 0.6 g of acid-washed quartz sand (20-40 mesh) to minimize temperature gradients, and the mixture was loaded into a stainless-steel fixed-bed reactor. 503 K, H₂O 0.3 mL/min, CH₄ 20 mL/min, CO 10 mL/min, O₂ 5 mL/min.

2. Catalytic promotion by acidic sites

To directly investigate the influence of acidic sites on catalytic performance, we

prepared a series of RhFe/ZSM-5-X catalysts with identical metal loadings but varying SiO₂/Al₂O₃ ratios (SiO₂/Al₂O₃ = 18, 25, 40, 60, 80). NH₃-TPD and pyridine adsorption infrared spectroscopy (Py-IR) results show that the acidity of the catalysts significantly decreases with increasing SiO₂/Al₂O₃ ratios (Fig. S25A-B).

Fig. S25. (a) NH₃-TPD with different SiO₂/Al₂O₃ ZSM-5 catalysts. (b) Infrared spectra of different SiO₂/Al₂O₃ catalysts. (c) The performance of different SiO₂/Al₂O₃ catalysts.

Catalytic performance tests (Fig. S25C) indicate that both the yield and selectivity of CH₃COOH decrease markedly as the support SiO₂/Al₂O₃ increases. Among these, RhFe/ZSM-5-18 exhibits the best CH₃COOH performance (18.2 mmol g_{cat}⁻¹ h⁻¹), while RhFe/ZSM-5-80 shows a substantial decline in activity (0.4 mmol g_{cat}⁻¹ h⁻¹). This clearly demonstrates that zeolite acidity is a key factor in maintaining high catalytic performance.

3. Further verification of acidic site function: Na⁺ exchange experiment

To further differentiate the acidic contribution, Na⁺ ion exchange was performed on RhFe/ZSM-5 (SiO₂/Al₂O₃=18) to obtain RhFe/Na-ZSM-5. Py-IR reveals a significant decrease in both Brønsted and Lewis acid sites, confirming effective neutralization of the acidic sites (Fig. S26A).

Fig. S26. (a) Infrared spectra of pyridine for RhFe/ZSM-5 and Na-RhFe/ZSM-5 catalysts. (b) The performance of RhFe/ZSM-5 and Na-RhFe/ZSM-5 catalysts. (c) CO-

TPD spectra.

Correspondingly, the CH₃COOH yield over RhFe/Na-ZSM-5 decreases to 3.8 mmol g_{cat}⁻¹ h⁻¹, further proving that acidic sites (particularly exchangeable H⁺) are indispensable for high catalytic activity Fig. S26B.

To investigate the mechanism behind the activity decline, we conducted CO-TPD analysis (Fig. S26C.). RhFe/ZSM-5 exhibits a strong CO desorption peak at 665 K, indicating its strong adsorption capacity for CO. In contrast, the desorption peak for RhFe/Na-ZSM-5 shifts to 553 K with reduced intensity, demonstrating that Na⁺ exchange not only weakens acidity but also the CO adsorption strength of catalyst. The reduced CO adsorption capacity may directly affect the formation of the •COOH intermediate and the subsequent C-C coupling, thereby lowering the efficiency of acetic acid synthesis.

In summary, the above experimental results consistently indicate that the acidic sites of the ZSM-5 support not only provide a necessary protonic chemical environment but also promote the formation of key intermediates by enhancing CO adsorption. These functions constitute the structural foundation for maintaining the high activity and selectivity of this catalytic system.

5. The EXAFS assignments of peaks to Fe-O-Fe and Fe-O-Rh are not fully convincing. For such assignments, reference samples with the proposed bonding environments are essential.

We thank the reviewer for this critical comment. We fully agree that having well-defined reference samples with Fe-O-Fe and Fe-O-Rh bonding environments would provide the most direct evidence for our assignment. Unfortunately, synthesizing molecular or solid-state compounds with well-characterized, isolated Fe-O-Rh units is extremely challenging, and despite our best efforts, we have not succeeded in obtaining the target compound.

Here, we wish to emphasize that our assignment of the Rh-O-Fe structure is not based solely on EXAFS fitting, but is robustly supported by a convergence of multiple independent characterization techniques and chemical evidence. To further strengthen our conclusion, we have supplemented the evidence with CO-IR spectroscopic data:

XPS and XANES Analysis (Electronic interaction):

Compared to Rh/ZSM-5, the Rh 3d XPS binding energy in RhFe/ZSM-5 shifts to a higher value (307.7 to 308.2 eV) (Fig. S23), while the Fe K-edge in XANES shifts to a lower energy (Fig. S22). This consistent trend provides strong evidence for electron transfer from Rh to Fe, which is a clear signature of a direct electronic interaction and the formation of Rh-O-Fe linkages, rather than separate Rh and Fe oxide domains.

Fig. S23. Electron transfer. (A) Rh 3d XPS spectra of Rh/ZSM-5 and FeRh/ZSM-5. **(B)** Fe 2p XPS spectra of Fe/ZSM-5 and FeRh/ZSM-5, respectively.

Fig. S22. (A) XANES spectra at the Rh K-edge of the Rh/ZSM-5, FeRh/ZSM-5, Rh₂O₃ and Rh foil. **(B)** XANES spectra at the Fe K-edge of the Fe/ZSM-5, FeRh/ZSM-5, FeO, Fe₂O₃ and Fe foil.

Spectroscopic verification of electronic structure via CO-DRIFTS

To further elucidate the unique electronic structure of the catalyst, we carried out in-situ CO-DRIFTS measurements (Fig. S24). The spectra of ZSM-5 and monometallic

Fe/ZSM-5 show characteristic peaks of gaseous CO at 2171 cm^{-1} and 2118 cm^{-1} , whereas Rh/ZSM-5 exhibits linear CO adsorption bands at 2107 cm^{-1} and 2040 cm^{-1} . Notably, on the RhFe/ZSM-5 catalyst, these bands undergo a distinct blue shift to 2104 cm^{-1} and 2037 cm^{-1} , indicating a decrease in the electron density of Rh sites. This directly reflects the electronic interaction between Rh and Fe, specifically, electron transfer from Rh to Fe, and is fully consistent with XPS and XANES data. In addition, the appearance of a bridged CO adsorption peak at 1860 cm^{-1} further confirms the presence of Rh nanoparticles, in agreement with HAADF-STEM observations.

Fig. S24. CO-IR adsorption spectrum.

Mössbauer Spectroscopy (environment of Fe):

The ^{57}Fe Mössbauer spectra measured at 4.2 K (Fig. 2E-F, Table S5) provide crucial complementary information. The significant increase in the proportion of dimeric $\text{Fe}^{\text{(III)}}$ species (from 18% in Fe/ZSM-5 to 28% in RhFe/ZSM-5) strongly

suggests that the introduction of Rh leads to the formation of new Fe-containing dimeric units.

Fig. 2. Structural characterization. (E-F) ^{57}Fe Mössbauer spectra for the Fe/ZSM-5 and RhFe/ZSM-5 catalysts.

Table S5. The Mössbauer fitted parameters of the Fe-zeolite samples, obtained at 4.2 K.

Sample/ Treatment	IS ($\text{mm}\cdot\text{s}^{-1}$)	QS ($\text{mm}\cdot\text{s}^{-1}$)	Hyperfine field (T)	Γ ($\text{mm}\cdot\text{s}^{-1}$)	Phase	Spectral contribution (%)
A. ^{57}Fe -ZSM-5	0.53	0.88	-	1.08	$\text{Fe}^{\text{III}}\text{-D}^{\text{a}}$	18
	0.50	0.03	41.6*	0.94	$\text{Fe}^{\text{III}}\text{-PHS}^{\text{b}}$	79
	1.38	3.37		0.72	$\text{Fe}^{\text{II}}^{\text{c}}$	3
B. Rh ^{57}Fe -ZSM-5	0.50	1.06	-	1.09	$\text{Fe}^{\text{III}}\text{-D}$	28
	0.50	0.03	41.9*	0.82	$\text{Fe}^{\text{III}}\text{-PHS}$	70
	1.32	3.51		0.72	Fe^{II}	2

Experimental uncertainties: Isomer shift: I.S. ± 0.02 mm s^{-1} ; Quadrupole splitting: Q.S. ± 0.05 mm s^{-1} ; Line width: $\Gamma \pm 0.05$ mm s^{-1} ; Hyperfine field: ± 0.2 T; Spectral contribution: $\pm 3\%$; *Average magnetic field; ^a Dimeric high-spin $\text{Fe}^{\text{III}}\text{-Fe}^{\text{III}}$ complexes; ^b Isolated (monomeric) Fe^{III} ions (paramagnetic hyperfine splitting); ^c Isolated Fe^{2+} ions.

Furthermore, compared to Fe/ZSM-5, this dimeric component in RhFe/ZSM-5 exhibits a lower isomer shift and altered quadrupole splitting, indicating that Fe is in a distinct electronic environment. We attribute this to the formation of mixed Fe-O-Rh dimers rather than pure Fe-O-Fe dimers. This interpretation is fully consistent with the

electron transfer observed by XPS/XANES.

Catalytic Performance and Kinetic Isotope Effect (Functional evidence):

The dramatic catalytic enhancement (5.7-fold yield increase, Table 1) and the distinct change in the rate-determining step (inverse KIE for RhFe/ZSM-5 vs. normal KIE for Rh/ZSM-5, Fig. 3A-B) were not observed with a physical mixture of Rh/ZSM-5 and Fe/ZSM-5. This provides compelling functional evidence for a synergistic Rh-Fe site that operates fundamentally differently from isolated Rh or Fe sites. The most plausible structural origin for this synergy is the formation of Rh-O-Fe pairs.

Table 1. Catalytic performance of RhFe/ZSM-5 catalysts for the oxidation of CH₄

		CH ₄ +O ₂ +CO \longrightarrow CH ₃ COOH							
Entry	Catalyst	T (K)	Reactants (MPa)			Productivity (mmol g _{cat} ⁻¹ h ⁻¹)			CH ₃ COOH Selectivity (%) ^d
			CH ₄	O ₂	CO	CH ₃ OH	HCOOH	CH ₃ COOH	
1	RhFe/ZSM-5	463	3	0.3	0.6	0.15	1.36	18.27	92
2	RhFe/ZSM-5	363	3	0.3	0.6	0.02	0.03	0.17	77
3	RhFe/ZSM-5	463	0	0.3	0.6	n.d. ^c	n.d. ^c	n.d. ^c	n.d. ^c
4	RhFe/ZSM-5	463	3	0	0.6	n.d. ^c	n.d. ^c	n.d. ^c	n.d. ^c
5	RhFe/ZSM-5	463	3	0.3	0	n.d. ^c	n.d. ^c	n.d. ^c	n.d. ^c
6	H-ZSM-5	463	3	0.3	0.6	n.d. ^c	n.d. ^c	n.d. ^c	n.d. ^c
7	Rh/ZSM-5	463	3	0.3	0.6	0.68	1.34	3.2	61
8	Fe/ZSM-5	463	3	0.3	0.6	n.d. ^c	n.d. ^c	n.d. ^c	n.d. ^c
9 ^a	Rh/ZSM-5//Fe/ZSM-5	463	3	0.3	0.6	0.71	1.42	3.64	63
10 ^b	Rh/ZSM-5-C	463	3	0.3	0.6	0.4	1.15	1.89	55
11 ^b	Fe/ZSM-5-C	463	3	0.3	0.6	n.d. ^c	n.d. ^c	n.d. ^c	n.d. ^c
12 ^b	RhFe/ZSM-5-C	463	3	0.3	0.6	3.8	1.58	5.08	49

Reaction conditions: catalyst (10 mg), water (20 mL), time (2 h), stirring speed 800 rotations per minute (rpm).

For entries 3-5 is the total pressure was maintained with N₂ or Ar. ^a Rh/ZSM-5 and Fe/ZSM-5 were physically

mixed. ^b ZSM-5-C was obtained from Nankai University Catalyst Co Ltd, which has the same metal loading

capacity and SiO₂: Al₂O₃ ratio (18) as RhFe/ZSM-5. ^c n.d., not detected. ^d CH₃COOH selectivity in liquid products.

Fig. 3. CH₄ adsorption and activation mechanism. (A) CH₃COOH production under H₂O and D₂O with different reaction time over Rh/ZSM-5. (B) CH₃COOH production under H₂O and D₂O with different reaction time over RhFe/ZSM-5.

We acknowledge the current lack of direct structural proof via a Fe-O-Rh reference sample. However, we are confident that the collective body of evidence from XPS, XANES, Mössbauer spectroscopy, and the unique catalytic behavior presents a very strong and self-consistent case for the existence of Rh-O-Fe linkages.

6. The respective roles of Rh and Fe remain ambiguous: the text suggests Rh(III) activates CH₄ while Fe^(IV)=O activates H₂O, yet Mössbauer data indicate Fe species are not strongly affected by CH₄. More clarification is needed.

We thank the reviewer for raising this important point. Indeed, the in situ Mössbauer data show that Fe^(IV)=O species are not significantly perturbed by CH₄ exposure (Fig. 4A-B, Treatment C), which aligns with our proposed decoupled activation mechanism. In this mechanism:

Fe(IV)=O is responsible for water activation, as evidenced by isotope-labeling experiments (Fig. 3E), which confirm that the hydroxyl oxygen in acetic acid originates exclusively from H₂O rather than O₂. Moreover, in-situ Mössbauer spectroscopy shows that Fe^(IV)=O species remain largely unaffected upon CH₄ introduction, but disappear

upon H₂O exposure, further corroborating the role of Fe^(IV)=O in water activation.

Rh^(III) is responsible for CH₄ activation, when CH₄ was introduced (Fig. 4A and B, Treatment C), the spectral contributions of Fe^(IV) species remained largely unchanged of RhFe/ZSM-5 in Mössbauer spectroscopy, but their hyperfine parameters of Fe^(IV) shifted more significantly than that Fe/ZSM-5, suggesting that the neighboring Rh sites play a crucial role in C-H activation, rather than the Fe^(IV)=O species. Upon CH₄ introduction to the RhFe/ZSM system post O₂/CO treatment, the HF-EPR signal re-emerged (Fig. 4C Treatment C), demonstrating that CH₄ can be activated by high-valent metals (Fe^(IV) or Rh^(III)). Importantly, the Mössbauer spectra revealed no significant change in Fe^(IV) species after CH₄ exposure, confirming that the observed signal in the HF-EPR originates from the reduction of Rh^(III) to Rh^(II), while Rh^(III) actively participated in CH₄ activation.

Fig. 3. (E) GC-MS spectra of the isotope CH₃COOH produced from CH₄ conversion when using CH₄+CO+¹⁶O₂+H₂¹⁸O, CH₄+CO+¹⁸O₂+H₂¹⁶O and CH₄+CO+O₂+D₂O as the reactants.

Fig. 4. *In-situ* Mössbauer spectroscopy. (A) Fe/ZSM-5. (B) RhFe/ZSM-5. Transmission ^{57}Fe Mössbauer spectra were collected at 4.2 K with a sinusoidal velocity spectrometer using a $^{57}\text{Co}(\text{Rh})$ source. Velocity calibration was carried out using an $\alpha\text{-Fe}$ foil at room temperature. The source and the absorbing samples were kept at the same temperature during the measurements. (C) *In-situ* High-field EPR of the catalyst during reaction. Reaction conditions: 463 K, 12 h, test conditions: 15 K, microwave frequency: 240 Hz.

Table S6. The Mössbauer fitted parameters of the Fe/ZSM-5 sample, obtained at 4.2 K.

Sample/ Treatment	IS ($\text{mm}\cdot\text{s}^{-1}$)	QS ($\text{mm}\cdot\text{s}^{-1}$)	Hyperfine field (T)	Γ ($\text{mm}\cdot\text{s}^{-1}$)	Phase	Spectral contribution (%)
A. 0.2% ^{57}Fe -ZSM-5 O_2/Ar 230 C, 20 bar, 2h	0.21	2.26	-	0.93	$\text{Fe}^{(\text{IV})}=\text{O}$ (I) cyan	26
	0.14	2.28	48.1	0.68	$\text{Fe}^{(\text{IV})}=\text{O}$ (II) magenta	10
	0.31	1.44	-	1.15	$\text{Fe}^{(\text{III})}\text{-D}^{\text{a}}$ red	30
	0.37	0.04	36.7*	0.70	$\text{Fe}^{(\text{III})}\text{-PHS}^{\text{b}}$ blue	30
	1.36	3.18	-	0.72	$\text{Fe}(\text{II})^{\text{c}}$ green	4
B. 0.2% ^{57}Fe -ZSM-5 $\text{O}_2/\text{CO}/\text{Ar}$ 230 C, 20 bar, 2h	0.18	2.13	-	0.88	$\text{Fe}^{(\text{IV})}=\text{O}$ (I)	19
	0.14	2.06	48.9	0.74	$\text{Fe}^{(\text{IV})}=\text{O}$ (II)	11
	0.36	1.52	-	1.32	$\text{Fe}^{(\text{III})}\text{-D}$	21
	0.38	0.03	37.4*	0.71	$\text{Fe}^{(\text{III})}\text{-PHS}$	34
	1.13	2.85	-	1.09	$\text{Fe}(\text{II})$	15

C.	0.19	2.06	-	0.98	Fe ^(IV) =O (I)	19
0.2% ⁵⁷ Fe-ZSM-5	0.13	2.08	49.1	0.73	Fe ^(IV) =O (II)	12
O ₂ /CO/CH ₄	0.35	1.32	-	1.14	Fe ^(III) -D	31
230 C, 20 bar, 2h	0.37	0.05	37.2*	0.72	Fe ^(III) -PHS	31
	1.29	2.92	-	0.84	Fe(II)	7
D.	0.37	1.39	-	0.90	Fe ^(III) -D	38
0.2% ⁵⁷ Fe-ZSM-5	0.35	0.03	42.1*	0.79	Fe ^(III) -PHS	46
O ₂ /CO/Ar/H ₂ O	1.29	2.70	-	0.98	Fe(II)	16
230 C, 20 bar, 2h						
E.	0.37	1.30	-	0.93	Fe ^(III) -D	41
0.2% ⁵⁷ Fe-ZSM-5	0.36	0.01	43.4*	0.74	Fe ^(III) -PHS	47
O ₂ /CO/CH ₄ /H ₂ O	1.35	2.70	-	0.96	Fe(II)	12
230 C, 20 bar, 2h						

Experimental uncertainties: Isomer shift: I.S. ± 0.02 mm s⁻¹; Quadrupole splitting: Q.S. ± 0.05 mm s⁻¹; Line width: $\Gamma \pm 0.05$ mm s⁻¹; Hyperfine field: ± 0.2 T; Spectral contribution: $\pm 3\%$; *Average magnetic field; ^aDimeric high-spin Fe^(III)-Fe^(III) or Fe^(III)-Fe^(IV) complexes; ^bIsolated (monomeric) Fe^(III) ions (paramagnetic hyperfine splitting); ^cIsolated Fe^(II) ions.

Table S7. The Mössbauer fitted parameters of the Fe(Rh)/ZSM-5 sample, obtained at 4.2 K.

Sample/ Treatment	IS (mm·s ⁻¹)	QS (mm·s ⁻¹)	Hyperfine field (T)	Γ (mm·s ⁻¹)	Phase	Spectral contribution (%)
A. Rh ⁵⁷ Fe-ZSM-5	0.21	2.27	-	0.82	Fe ^(IV) =O (I)	16
O ₂ /Ar	0.16	2.21	48.3	0.65	Fe ^(IV) =O (II)	10
230 C, 20 bar, 2h	0.31	1.50	-	1.13	Fe ^(III) -D ^a	43
	0.35	0.03	36.4*	0.75	Fe ^(III) -PHS ^b	29
	1.36	3.23	-	0.74	Fe(II) ^c	2
B. 0.6% Rh	0.22	2.24	-	0.82	Fe ^(IV) =O (I)	18
0.2% ⁵⁷ Fe-ZSM-5	0.18	2.22	48.3	0.55	Fe ^(IV) =O (II)	9
O ₂ /CO/Ar	0.32	1.61	-	1.26	Fe ^(III) -D	31
230 C, 20 bar, 2h	0.32	0.04	37.2*	0.75	Fe ^(III) -PHS	35
	1.26	3.09	-	0.99	Fe(II)	7
C. Rh ⁵⁷ Fe-ZSM-5	0.17	1.88	-	1.20	Fe ^(IV) =O (I)	19
O ₂ /CO/CH ₄	0.17	1.98	49.2	0.70	Fe ^(IV) =O (II)	8
230 C, 20 bar, 2h	0.37	1.36	-	0.98	Fe ^(III) -D	29
	0.38	0.04	38.6*	0.75	Fe ^(III) -PHS	32
	1.32	2.81	-	0.99	Fe(II)	12

D. Rh ⁵⁷ Fe-ZSM-5	0.34	1.52	-	1.32	Fe ^(III) -D	27
O ₂ /CO/Ar/H ₂ O	0.33	0.01	40.9*	0.97	Fe ^(III) -PHS	70
230 C, 20 bar, 2h	1.32	3.18	-	0.74	Fe(II)	3
E. Rh ⁵⁷ Fe-ZSM-5	0.32	1.54	-	1.12	Fe ^(III) -D	26
O ₂ /CO/CH ₄ /H ₂ O	0.36	0.01	41.5*	1.09	Fe ^(III) -PHS	57
230 C, 20 bar, 2h	1.32	2.91	-	1.02	Fe(II)	17

Experimental uncertainties: Isomer shift: I.S. ± 0.02 mm s⁻¹; Quadrupole splitting: Q.S. ± 0.05 mm s⁻¹; Line width: $\Gamma \pm 0.05$ mm s⁻¹; Hyperfine field: ± 0.2 T; Spectral contribution: $\pm 3\%$; *Average magnetic field; ^aDimeric high-spin Fe^(III)-Fe^(III) or Fe^(III)-Fe^(IV) complexes; ^bIsolated (monomeric) Fe^(III) ions (paramagnetic hyperfine splitting); ^cIsolated Fe^(II) ions.

7. No direct detection of CH₃ or COOH radicals intermediates is shown by EPR.

Radical trapping experiments are necessary to confirm the presence of these species.

We thank the reviewer for the valuable suggestion. To directly detect the radical intermediates during the reaction, we have conducted in-situ spin-trapping experiments using 5,5-dimethyl-1-pyrroline N-oxide (DMPO) under standard reaction conditions (CH₄/CO/O₂/H₂O). The experiment successfully captured radical signals associated with •OH and •CH₃, further confirming that the methyl radical is the key intermediate in methane activation within this system (Fig. S39).

Fig. S39. *In-situ* EPR spectra under CH₄+CO+O₂+H₂O.

Although •COOH signals were not directly observed in the in-situ EPR, we performed supplementary verification of this intermediate using in-situ infrared

spectroscopy. According to the proposed mechanism, if $\bullet\text{COOH}$ is formed via the coupling of $\bullet\text{OH}$ and CO on Fe sites, the corresponding intermediate signal should be detectable on the Fe/ZSM-5 catalyst. To this end, we conducted in-situ infrared tests on Fe/ZSM-5 under an $\text{O}_2/\text{CO}/\text{H}_2\text{O}$ atmosphere and successfully observed characteristic vibrational peaks at 1576 cm^{-1} and 1352 cm^{-1} , which are assigned to the $\bullet\text{COOH}$ intermediate. This result further confirms that $\bullet\text{COOH}$ is a key reaction intermediate in the acetic acid synthesis pathway.

The relevant content has been added to the manuscript in Supplementary Figure S39-40.

Fig. S40. *In-situ* diffuse reflectance infrared fourier transform spectra collected at the $\text{H}_2\text{O} + \text{O}_2$ and $\text{O}_2/\text{H}_2\text{O}/\text{CO}$ at 463 K over Fe/ZSM-5.

Reviewer #2 (Remarks to the Author):

Typical metals like Rh and Fe have been widely reported for methane oxidative carbonylation to produce acetic acid in both CH₄/CO/O₂ and CH₄/CO/H₂O₂ systems. Rh-Fe dual-atom sites on MoS₂ and Ir-O-Ir dimeric sites on SBA-15 have achieved high acetic acid selectivity at low temperatures (J. Am. Chem. Soc. 2025, 147, 14530; J. Am. Chem. Soc. 2023, 145, 769). Here, the authors report Rh-O-Fe dual-metal sites that enable CH₃COOH synthesis at higher temperatures (463 K) with high yield and selectivity, typically shows low catalytic activity below 423 K. Unlike the conventional CO insertion into M-CH₃, they propose a direct carbon-carbon coupling mechanism, offering a more feasible pathway for acetic acid formation, similar pathway has also been reported (Appl. Catal. B 2025, 378, 125632). Although the findings sound good and the manuscript is meticulously prepared, the overall novelty of this study appears constrained by prior publications in this field. Additionally, several substantive concerns preclude my endorsement for publication in Nature Communications in its current form.

1. The assertion of Rh-O-Fe structure formation relies primarily on indirect evidence from Fe/Rh K-edge FT-EXAFS and ⁵⁷Fe Mössbauer spectroscopic analyses. However, these techniques cannot provide unambiguous structural confirmation. To strengthen their claim, the authors should incorporate direct visualization through aberration-corrected HAADF-STEM and additional spectroscopic validation via CO-IR measurements.

We sincerely appreciate the reviewer's critical suggestions, which are essential for strengthening the evidence regarding the Rh-O-Fe structure. We agree that direct visual observation and supplementary spectroscopic verification would significantly enhance the persuasiveness of the conclusions. Following these recommendations, we have conducted the relevant experiments and present new experimental results below, which collectively provide robust multi-faceted evidence supporting the existence of Rh-O-Fe linkages.

1. Spectroscopic verification of electronic structure via CO-DRIFTS

To further elucidate the unique electronic structure of the catalyst, we carried out

in-situ CO-DRIFTS measurements (Fig. S24). The spectra of ZSM-5 and monometallic Fe/ZSM-5 show characteristic peaks of gaseous CO at 2071 cm^{-1} and 2118 cm^{-1} , whereas Rh/ZSM-5 exhibits linear CO adsorption bands at 2107 cm^{-1} and 2040 cm^{-1} . Notably, on the RhFe/ZSM-5 catalyst, these bands undergo a distinct blue shift to 2104 cm^{-1} and 2037 cm^{-1} , indicating a decrease in the electron density of Rh sites. This directly reflects the electronic interaction between Rh and Fe, specifically, electron transfer from Rh to Fe, and is fully consistent with XPS and XANES data. In addition, the appearance of a bridged CO adsorption peak at 1860 cm^{-1} further confirms the presence of Rh nanoparticles, in agreement with HAADF-STEM observations.

Fig. S24. CO-IR adsorption spectrum.

2. Direct Structural Observation via Aberration-Corrected HAADF-STEM

We performed detailed AC-HAADF-STEM analysis. High-resolution images (Fig. S17) clearly show the microporous structure of ZSM-5, dispersed nanoparticles,

and single atoms distributed adjacently in multiple regions. By combining the results from CO-IR and EXAFS spectra (Fig. S24, Fig. 2C-D), the existence of Rh nanoparticles and Fe single atoms has been demonstrated. These results directly reveal the spatial proximity between Rh and Fe, corroborating Mössbauer spectroscopy data, and providing visual morphological evidence for the Rh-O-Fe coordination structure (Fig. 2E-F).

Fig. S17. AC-HAADF-STEM of RhFe/ZSM-5.

In summary, AC-HAADF-STEM imaging provides direct spatial evidence of Rh-O-Fe pairing at the atomic scale, while CO-DRIFTS data reveal the consequent modification of the electronic structure of Rh sites at the spectroscopic level. These new experimental results, together with the previously presented EXAFS and Mössbauer data, form a coherent and self-consistent body of evidence that clearly confirms the formation of the Rh-O-Fe structure in our catalytic system.

We have integrated these new findings and related discussions into the revised manuscript. Once again, we sincerely thank the reviewer for their valuable suggestions, which have significantly enhanced the reliability and completeness of the key structural arguments in this work.

2. The authors propose a reaction mechanism for CH₃COOH formation involving direct carbon-carbon coupling between ·CH₃ and ·COOH radicals, same as previous study (Angew. Chem. Int. Ed. 2024, 63, e202315343; Appl. Catal. B 2025, 378, 125632). However, this claim requires more substantial evidence. First, the manuscript fails to provide direct experimental evidence for ·CH₃ radical capture (EPR). Second, while *COOH intermediates are known to form in CO/O₂/H₂O systems (ACS Catal.

2020, 10, 13993). From Fig. 3C, the formation of H₂O₂ is clearly observed, indicating that Rh sites can indeed catalyze the reaction of CO, O₂, and H₂O to produce H₂O₂ via *COOH intermediates. However, Fig. 5B presents a strikingly distinct O₂ activation pathway on Rh sites for methane oxidation. The authors should provide more evidence about this phenomenon.

We sincerely thank the reviewer for these insightful and constructive comments, which have enabled us to significantly strengthen the evidence for our proposed mechanism and clarify its novelty. We have conducted additional experiments to address these issues.

1. Direct experimental evidence for ·CH₃ radicals

We agree that direct detection of radical intermediates is crucial for validating the proposed mechanism. Following the reviewer's suggestion, we performed electron paramagnetic resonance experiments using the spin trap DMPO. When the RhFe/ZSM-5 catalyst in a DMPO-containing suspension was exposed to reaction atmosphere, we observed a characteristic EPR signal (Fig. S39). This signal matches the data reported in the literature for the DMPO-·CH₃ and DMPO-OH adduct. This provides direct and unambiguous evidence for the generation of ·CH₃ radicals from CH₄ activation on our catalyst.

Fig. S39. *In-situ* EPR spectra under CH₄+CO+O₂+H₂O.

2. Elucidating the unique O₂ activation pathway and addressing the H₂O₂ observation.

The reviewer correctly notes that H₂O₂ formation is observed on monometallic Rh/ZSM-5 (Fig. 3C), indicating H₂O₂ as the intermediate product pathway. However, we wish to emphasize that our proposed unique pathway (Fig. 5B) relies precisely on the synergistic effect of the Rh-O-Fe dual sites, rather than isolated Rh sites. The key evidence is as follows:

Crucial difference in the dual-site system: in the bimetallic RhFe/ZSM-5 catalyst (Fig. 3D), we did not detect H₂O₂. This stark contrast with monometallic Rh/ZSM-5 (Fig. 3C) indicates a fundamental change in the reaction pathway within RhFe/ZSM-5.

Fig. 3. (C-D) Detection of H₂O₂ using *in-situ* DRIFTS spectra of Rh/ZSM-5 and FeRh/ZSM-5 catalysts.

Dominance of the Fe^(IV)=O-mediated pathway: we propose that in RhFe/ZSM-5, O₂ preferentially oxidizes Fe sites to highly reactive Fe^(IV)=O species (confirmed by Mössbauer spectroscopy, Fig. 4 and Table S6, S7), which can directly cleave H₂O to generate ·OH, bypassing the multi-step pathway via H₂O₂ formation.

Fig. 4. *In-situ* Mössbauer spectroscopy. (A) Fe/ZSM-5. (B) RhFe/ZSM-5. Transmission ^{57}Fe Mössbauer spectra were collected at 4.2 K with a sinusoidal velocity spectrometer using a $^{57}\text{Co}(\text{Rh})$ source. Velocity calibration was carried out using an α -Fe foil at room temperature. The source and the absorbing samples were kept at the same temperature during the measurements.

Table S6. The Mössbauer fitted parameters of the Fe/ZSM-5 sample, obtained at 4.2 K.

Sample/ Treatment	IS ($\text{mm}\cdot\text{s}^{-1}$)	QS ($\text{mm}\cdot\text{s}^{-1}$)	Hyperfine field (T)	Γ ($\text{mm}\cdot\text{s}^{-1}$)	Phase	Spectral contribution (%)
A. 0.2% ^{57}Fe -ZSM-5 O_2/Ar 230 C, 20 bar, 2h	0.21	2.26	-	0.93	$\text{Fe}^{(\text{IV})}=\text{O}$ (I) cyan	26
	0.14	2.28	48.1	0.68	$\text{Fe}^{(\text{IV})}=\text{O}$ (II) magenta	10
	0.31	1.44	-	1.15	$\text{Fe}^{(\text{III})}$ -D ^a red	30
	0.37	0.04	36.7*	0.70	$\text{Fe}^{(\text{III})}$ -PHS ^b blue	30
	1.36	3.18	-	0.72	$\text{Fe}(\text{II})^c$ green	4
B. 0.2% ^{57}Fe -ZSM-5 $\text{O}_2/\text{CO}/\text{Ar}$ 230 C, 20 bar, 2h	0.18	2.13	-	0.88	$\text{Fe}^{(\text{IV})}=\text{O}$ (I)	19
	0.14	2.06	48.9	0.74	$\text{Fe}^{(\text{IV})}=\text{O}$ (II)	11
	0.36	1.52	-	1.32	$\text{Fe}^{(\text{III})}$ -D	21
	0.38	0.03	37.4*	0.71	$\text{Fe}^{(\text{III})}$ -PHS	34
	1.13	2.85	-	1.09	$\text{Fe}(\text{II})$	15
C. 0.2% ^{57}Fe -ZSM-5 $\text{O}_2/\text{CO}/\text{CH}_4$ 230 C, 20 bar, 2h	0.19	2.06	-	0.98	$\text{Fe}^{(\text{IV})}=\text{O}$ (I)	19
	0.13	2.08	49.1	0.73	$\text{Fe}^{(\text{IV})}=\text{O}$ (II)	12
	0.35	1.32	-	1.14	$\text{Fe}^{(\text{III})}$ -D	31
	0.37	0.05	37.2*	0.72	$\text{Fe}^{(\text{III})}$ -PHS	31

	1.29	2.92	-	0.84	Fe(II)	7
D.	0.37	1.39	-	0.90	Fe ^(III) -D	38
0.2% ⁵⁷ Fe-ZSM-5	0.35	0.03	42.1*	0.79	Fe ^(III) -PHS	46
O ₂ /CO/Ar/H ₂ O	1.29	2.70	-	0.98	Fe(II)	16
230 C, 20 bar, 2h						
E.	0.37	1.30	-	0.93	Fe ^(III) -D	41
0.2% ⁵⁷ Fe-ZSM-5	0.36	0.01	43.4*	0.74	Fe ^(III) -PHS	47
O ₂ /CO/CH ₄ /H ₂ O	1.35	2.70	-	0.96	Fe(II)	12
230 C, 20 bar, 2h						

Experimental uncertainties: Isomer shift: I.S. ± 0.02 mm s⁻¹; Quadrupole splitting: Q.S. ± 0.05 mm s⁻¹; Line width: $\Gamma \pm 0.05$ mm s⁻¹; Hyperfine field: ± 0.2 T; Spectral contribution: $\pm 3\%$; *Average magnetic field; ^aDimeric high-spin Fe^(III)-Fe^(III) or Fe^(III)-Fe^(IV) complexes; ^bIsolated (monomeric) Fe^(III) ions (paramagnetic hyperfine splitting); ^cIsolated Fe^(II) ions.

Table S7. The Mössbauer fitted parameters of the Fe(Rh)/ZSM-5 sample, obtained at 4.2 K.

Sample/ Treatment	IS (mm·s ⁻¹)	QS (mm·s ⁻¹)	Hyperfine field (T)	Γ (mm·s ⁻¹)	Phase	Spectral contribution (%)
A. Rh ⁵⁷ Fe-ZSM-5	0.21	2.27	-	0.82	Fe ^(IV) =O (I)	16
O ₂ /Ar	0.16	2.21	48.3	0.65	Fe ^(IV) =O (II)	10
230 C, 20 bar, 2h	0.31	1.50	-	1.13	Fe ^(III) -D ^a	43
	0.35	0.03	36.4*	0.75	Fe ^(III) -PHS ^b	29
	1.36	3.23	-	0.74	Fe(II) ^c	2
B. 0.6% Rh	0.22	2.24	-	0.82	Fe ^(IV) =O (I)	18
0.2% ⁵⁷ Fe-ZSM-5	0.18	2.22	48.3	0.55	Fe ^(IV) =O (II)	9
O ₂ /CO/Ar	0.32	1.61	-	1.26	Fe ^(III) -D	31
230 C, 20 bar, 2h	0.32	0.04	37.2*	0.75	Fe ^(III) -PHS	35
	1.26	3.09	-	0.99	Fe(II)	7
C. Rh ⁵⁷ Fe-ZSM-5	0.17	1.88	-	1.20	Fe ^(IV) =O (I)	19
O ₂ /CO/CH ₄	0.17	1.98	49.2	0.70	Fe ^(IV) =O (II)	8
230 C, 20 bar, 2h	0.37	1.36	-	0.98	Fe ^(III) -D	29
	0.38	0.04	38.6*	0.75	Fe ^(III) -PHS	32
	1.32	2.81	-	0.99	Fe(II)	12

D. Rh ⁵⁷ Fe-ZSM-5	0.34	1.52	-	1.32	Fe ^(III) -D	27
O ₂ /CO/Ar/H ₂ O	0.33	0.01	40.9*	0.97	Fe ^(III) -PHS	70
230 C, 20 bar, 2h	1.32	3.18	-	0.74	Fe(II)	3
E. Rh ⁵⁷ Fe-ZSM-5	0.32	1.54	-	1.12	Fe ^(III) -D	26
O ₂ /CO/CH ₄ /H ₂ O	0.36	0.01	41.5*	1.09	Fe ^(III) -PHS	⁵⁷
230 C, 20 bar, 2h	1.32	2.91	-	1.02	Fe(II)	17

Experimental uncertainties: Isomer shift: I.S. ± 0.02 mm s⁻¹; Quadrupole splitting: Q.S. ± 0.05 mm s⁻¹; Line width: $\Gamma \pm 0.05$ mm s⁻¹; Hyperfine field: ± 0.2 T; Spectral contribution: $\pm 3\%$; *Average magnetic field; ^aDimeric high-spin Fe^(III)-Fe^(III) or Fe^(III)-Fe^(IV) complexes; ^bIsolated (monomeric) Fe^(III) ions (paramagnetic hyperfine splitting); ^cIsolated Fe^(II) ions.

This pathway difference is directly reflected in the catalytic performance: the acetic acid production rate of RhFe/ZSM-5 (18.2 mmol g⁻¹ h⁻¹) is much higher than that of Rh/ZSM-5 (3.2 mmol g⁻¹ h⁻¹), and it exhibits a unique inverse KIE (Fig. 3B), indicating that water activation is no longer the rate-determining step.

Fig. 3. CH₄ adsorption and activation mechanism. (A) CH₃COOH production under H₂O and D₂O with different reaction time over Rh/ZSM-5. (B) CH₃COOH production under H₂O and D₂O with different reaction time over RhFe/ZSM-5.

In summary, while the monometallic Rh/ZSM-5 indeed follows the traditional, less efficient pathway involving H₂O₂, the construction of the Rh-O-Fe dual site opens a new, more efficient route where O₂ is used to generate Fe^(IV)=O, enabling direct water activation for selective carbonylation. We have revised the discussion in the manuscript to articulate this contrast and its underlying reasons more clearly.

Fig. 5. (B) Proposed RhFe/ZSM-5 catalytic cycle.

3. The coupling of *COOH with *CH_3 species to produce acetic acid needs verification. As a proof of concept, if CH_3I (provided CH_3 radicals) were added as substrate, significantly enhanced acetic acid production would be expected. Furthermore, following the proposed mechanism, using ethane as substrate should yield substantial amounts of CH_3CH_2COOH .

Fig. S42. The liquid products after adding CH_3I .

We sincerely appreciate the reviewer's insightful and constructive comments. In response, we have conducted additional experiments involving the introduction of CH_3I . Under the optimized reaction conditions ($CO/O_2/H_2O$, RhFe/ZSM-5, 463 K), the addition of 200 μL CH_3I resulted in a 2.4-fold increase in CH_3COOH productivity compared to the system without CH_3I (Fig. S42). Notably, methanol formation was also

significantly enhanced ($38.6 \text{ mmol g}^{-1} \text{ h}^{-1}$). These results indicate that the externally supplied $\cdot\text{CH}_3$ radicals can not only directly couple with $\cdot\text{OH}$ to form CH_3OH , but also efficiently couple with $\cdot\text{COOH}$ intermediates to produce acetic acid, which is consistent with the proposed reaction pathway.

Fig. S15. ^1H NMR of CH_3CH_3 after reaction.

Synthesis of propionic acid using ethane as the substrate

Experiments using ethane as the substrate further validate the reaction mechanism (Fig. S15). Under conditions where ethane replaced methane ($\text{C}_2\text{H}_6/\text{CO}/\text{O}_2/\text{H}_2\text{O}$, RhFe/ZSM-5, 463 K), the formation of both acetic acid and propionic acid was detected in the reaction system. This result confirms that ethane can generate acetic acid via direct coupling of $\cdot\text{CH}_2\text{CH}_3$ with $\cdot\text{OH}$, while propionic acid is formed through coupling of $\cdot\text{CH}_2\text{CH}_3$ with $\cdot\text{COOH}$ radicals. These findings further support the proposed pathway in the manuscript, in which methane is converted to acetic acid via coupling of $\cdot\text{CH}_3$ with $\cdot\text{COOH}$. Additionally, the detection of C_1 oxygenates indicates that C-C bond cleavage of ethane may also occur under the applied reaction conditions.

4. The authors propose that the in situ formation of high-valent $\text{Fe}^{(\text{IV})}=\text{O}$ initiates a truncated WGS pathway, wherein H_2O is directly dissociated into $\cdot\text{OH}$. These radicals rapidly react with CO to form $\cdot\text{COOH}$ intermediates, which then couple with $\cdot\text{CH}_3$ to

yield CH_3COOH . To validate this mechanism, in situ DRIFTS studies on Fe/ZSM-5 are recommended: Upon sequential exposure to O_2 and CO, the formation of $\cdot\text{COOH}$ intermediates should be observable.

We sincerely appreciate the reviewer's insightful suggestions regarding the in situ spectroscopic validation of the proposed reaction pathway. In response to your recommendations, we performed systematic in situ DRIFTS experiments on Fe/ZSM-5 to directly capture the formation of the $\cdot\text{COOH}$ intermediate under sequential exposure to O_2 and CO. The results provide strong support for the proposed mechanism.

To explicitly verify whether $\text{Fe}^{(\text{IV})}=\text{O}$ promotes the direct cleavage of H_2O into $\cdot\text{OH}$ and subsequently facilitates CO hydroxylation to form $\cdot\text{COOH}$, we carried out stepwise in situ DRIFTS measurements under controlled conditions (Fig. S40). The experimental procedure was as follows: First, water vapor was introduced into the system via bubbling in a nitrogen stream, and a stable baseline spectrum was recorded. Subsequently, oxygen and carbon monoxide were introduced sequentially while the spectral changes were monitored in real time.

The experimental results show that no significant spectral changes were observed upon the introduction of oxygen. However, after CO was introduced, distinct CO adsorption peaks appeared at 2170 cm^{-1} and 2119 cm^{-1} . More importantly, characteristic vibrational peaks corresponding to the $\cdot\text{COOH}$ species were simultaneously observed at 1576 cm^{-1} and 1352 cm^{-1} . This result directly confirms the pathway involving the coupling of $\cdot\text{OH}$ with CO to form $\cdot\text{COOH}$, further supporting the reaction mechanism proposed in this work, in which the $\cdot\text{COOH}$ intermediate plays a key role.

The relevant spectral data have been updated in the (Fig. S40) and are referenced and discussed in the corresponding section of the manuscript.

Fig. S40. *In-situ* diffuse reflectance infrared fourier transform spectra collected at the H₂O +O₂ and O₂/H₂O/CO at 463 K over Fe/ZSM-5.

Reviewer #3 (Remarks to the Author):

In this article, Zhang and coauthors presented Rh-Fe site-supported ZSM-5 towards selective CH₄ conversion to CH₃COOH. The well-designed RhFe/ZSM-5 exhibits quite high CH₃COOH production rate 18.2 mmol g_{cat}⁻¹ h⁻¹, with 92% selectivity, which is much competitive among the current promising catalysts. Great efforts have been devoted to the study on the characterization and involved catalytic conversion process by in-situ Mössbauer spectroscopy, in-situ high-field EPR and in-situ FT-IR, etc. In particular, the mechanism of the •COOH and •CH₃ to yield CH₃COOH under Rh^(III)/Fe^(IV)=O active site was revealed. Overall, the conclusion is supported by solid evidence from both comprehensive experimental results and theoretical calculation, with adequate novelty for the material designing, synthesis and insights validation. Hence, I would like to recommend this work to be published after minor revision.

1. This paper presents a sustainable methane conversion strategy. By comparing it with existing approaches, the work highlights the advantages of the proposed catalytic method, thereby strengthening the manuscript's overall contribution.

Response:

The methane one-step oxidation process for acetic acid synthesis demonstrates significant advantages in terms of direct feedstock utilization, mild reaction conditions, process simplicity, and environmental friendliness. Using methane, carbon monoxide, and oxygen as raw materials, this process directly produces acetic acid in an aqueous phase at 190°C, thereby bypassing multiple energy-intensive steps required in the conventional methanol carbonylation route, such as syngas production and methanol synthesis. This not only greatly simplifies the process chain and reduces equipment investment and energy consumption but also avoids the use of corrosive promoters like CH₃I, minimizing equipment corrosion and environmental pollution. Moreover, the mild reaction conditions ensure safer and more controllable operations, while the aqueous-phase system aligns better with green chemistry principles. Although this technology is still under development and requires further optimization in terms of conversion efficiency, catalyst cost, and process scale-up, it holds clear application

prospects for the valorization of distributed resources such as natural gas and biogas, offering a potentially more atom-economical and sustainable pathway for acetic acid production.

Based on the reviewers' suggestions, we have analyzed the advantages of current methane conversion technologies and the limitations of methanol carbonylation for acetic acid production. In the revised manuscript, we have supplemented the detailed discussion as follows: The one-step CH₄ oxidation process for CH₃COOH production enables efficient synthesis through direct methane activation under mild conditions, bypassing the complex steps required in traditional methanol carbonylation processes, such as methanol synthesis and the use of halogen promoters, thus demonstrating significant advantages in feedstock economy and process simplicity. Although this new pathway still requires further refinement, it has already provided a more atom efficient and sustainable alternative for the green production of CH₃COOH.

2. The notation for Fe^(IV)/Fe⁴⁺ and Rh^(III)/Rh³⁺ should be consistent throughout the text.

Response: We thank the reviewer for this careful observation regarding the consistency of our notation. We sincerely apologize for this oversight. We agree that maintaining consistent notation for oxidation states is crucial for clarity and scientific rigor throughout the manuscript.

In response to this comment, we have carefully revised the entire text to adopt a consistent format for representing oxidation states. Specifically, we have standardized the notation to use Roman numerals in parentheses, such as Fe^(IV) and Rh^(III), instead of the mixed use of Roman numerals and Arabic numbers with plus signs (e.g., Fe⁴⁺, Rh³⁺). All other metal oxidation states (e.g., Fe^(III), Fe^(II)) have been adjusted to this format accordingly.

This change has been implemented across all relevant sections, including the Abstract, Introduction, Results, Discussion, and Conclusion.

We believe this revision ensures consistency and improves the readability of the manuscript. We have carefully checked the entire text to confirm that all oxidation state notations now follow this unified style.

Thank you again for helping us improve the quality of our manuscript.

3. Through extensive comparative experiments, the authors provide a detailed demonstration of the RhFe/ZSM-5 catalyst's effectiveness. Beyond the comparison of different metals, have the authors considered evaluating other molecular sieves, such as MOR, SSZ-13, or SiO₂?

Response:

Fig. S7. Catalytic performance over different supports, reaction conditions: 463 K, 10 mg catalyst, 2 h, 20 mL H₂O, 40 bar CH₄, 3 bar O₂, 6 bar CO.

We sincerely thank the reviewer for this insightful comment. Your suggestion to evaluate different supports is highly valuable, as it indeed helps to more comprehensively reveal the relationship between catalyst structure and performance. We have compared the performance of the RhFe catalyst supported on various carriers, including MOR, SSZ-13, and SiO₂. The detailed catalytic data have been compiled in the supplementary figure (Fig. S7).

The results demonstrate that RhFe/ZSM-5 exhibits significantly superior catalytic activity and selectivity compared to the other supports. This phenomenon is primarily attributed to the unique structural characteristics of ZSM-5: firstly, its suitable pore structure provides an optimal confined environment for the reaction intermediates, effectively promoting the C-C coupling between •CH₃ and •COOH radicals; secondly, the acidic nature of its framework effectively stabilizes the atomically dispersed Rh-O-Fe active sites while avoiding the initiation of undesirable side reactions.

These comparative experiments further confirm the unique advantages of ZSM-5

as a support in our reaction system and highlight the sophisticated synergistic effect between our designed Rh-O-Fe active center and the ZSM-5 carrier.

4. The authors evaluated the catalyst's performance under pressurized conditions. However, further studies under ambient conditions would be beneficial, as they facilitate easier scaling and mitigate the safety concerns of handling high-pressure methane-oxygen mixtures.

Response:

Fig. S4. (A) Catalytic performance at different pressure, reaction conditions: 10 mg catalyst, 2 h, 20 mL H₂O, CH₄/O₂/CO=10/1/2.

We thank the reviewer for raising this critical point regarding process safety and potential scalability. We fully agree that operation under ambient pressure is highly desirable from both practical and safety perspectives. Following this suggestion, we have conducted additional experiments to evaluate the catalytic performance under atmospheric pressure.

As shown in Fig. S4A, when the reaction was carried out at atmospheric pressure, the RhFe/ZSM-5 catalyst still demonstrated measurable activity, producing acetic acid at a rate of 0.24 mmol g_{cat.}⁻¹ h⁻¹. This confirms that the catalyst remains fundamentally active even without applied pressure, though the productivity is significantly lower than that under optimized pressurized conditions (18.2 mmol g_{cat.}⁻¹ h⁻¹). This difference is primarily attributed to the drastic reduction in the dissolved concentration of the gaseous reactants (CH₄ and CO) in the aqueous phase at ambient pressure, as governed by Henry's law. The inherently low concentration of reactants at the catalyst surface

limits the reaction rate.

5. The authors convincingly demonstrated the existence of an Rh-O-Fe structure in the catalyst through EXAFS and ^{57}Fe Mössbauer spectra. Have they considered how the catalytic performance would be affected if the Rh-O-Fe structure were absent, or if Rh and Fe were spatially separated? Such an investigation could further corroborate the crucial role of the Rh-O-Fe structure in the oxidative coupling of methane, which would be a very interesting demonstration.

Response:

We sincerely thank the reviewer for their insightful comments. The suggestion to validate the crucial role of the Rh-O-Fe structure through comparative experiments is highly constructive and directly addresses the core of our mechanistic investigation.

To systematically explore this issue, we specifically designed and prepared a comparative catalyst with spatially separated active sites: Fe/ZSM-5 was first synthesized via the in-situ seed method, followed by the introduction of Rh species through impregnation and subsequent calcination at 550°C , yielding a catalyst with spatially separated Rh/Fe sites (denoted as RhFe/ZSM-I).

Fig. S2B. The performance of different metal in ZSM-5.

The physically mixed Rh/ZSM-5+Fe/ZSM-5 system showed an acetic acid production rate of only $3.6 \text{ mmol g}_{\text{cat}}^{-1} \text{ h}^{-1}$, comparable to the activity of monometallic Rh/ZSM-5 alone. The impregnated RhFe/ZSM-I catalyst exhibited an improved production rate of $6.7 \text{ mmol g}_{\text{cat}}^{-1} \text{ h}^{-1}$. In contrast, the closely coupled Rh-O-Fe structure constructed in this work achieved the highest production rate of $18.2 \text{ mmol g}_{\text{cat}}^{-1} \text{ h}^{-1}$.

This gradient in activity is highly convincing: when Rh and Fe are spatially separated (physical mixture), the synergistic effect completely disappears; even when partial contact is achieved via impregnation (RhFe/ZSM-I), the activity remains limited; optimal catalytic performance is only achieved when the two elements form an atomically close Rh-O-Fe structure, enabling efficient coupling of $\cdot\text{CH}_3$ and $\cdot\text{COOH}$ radicals.

Dear Reviewers,

We sincerely thank the reviewer for the insightful and constructive comments. We acknowledge the importance of addressing the formation of CO₂ in our catalytic system and agree that the text should accurately reflect the product distribution. Below we provide our response and the corresponding revisions made to the manuscript.

Reviewer #1:

The authors have addressed almost all of my comments. However, it is clear that, beyond novelty, a significant issue is the substantial formation of CO₂ during the reaction of CO oxidation, reaching up to 50 mmol g⁻¹ h⁻¹, which is even higher than the amount of acetic acid produced. This point must be addressed in the manuscript before acceptance. The amount of CO₂ formed is almost twice that of acetic acid, and the text should therefore be corrected, in particular the statement: "In addition, the similar amount of CO₂ generated...". The authors should report the selectivity to acetic acid either by explicitly accounting for CO₂ formation or by clearly distinguishing between liquid phase selectivity and overall selectivity including CO₂.

Response:

We thank the reviewer for highlighting this important point. We agree that the substantial CO₂ formation should be clearly discussed and that the wording regarding CO₂ selectivity should be clarified. We have now revised the manuscript as follows:

In addition, comparative experiments performed with CH₃COOH in the presence and absence of CO confirmed that CO₂ is predominantly formed through CO oxidation (Fig. S11b). Under standard methane oxidation conditions, the rate of CO₂ production reached 41.6 mmol g_{cat}⁻¹ h⁻¹, surpassing that from CH₃COOH. For clarity, both liquid-phase selectivity (>92% toward CH₃COOH among oxygenated products) and total carbon selectivity (including CO₂, 42.3% for CH₃COOH) are provided (Fig. S12-S13). These results indicate that, although the catalyst shows high selectivity to acetic acid in the liquid phase, the oxidation of CO to CO₂ represents a major competing pathway and warrants further optimization in future studies.

Fig. S11. (b) The yields of CO₂ when feed acetic acid into the reaction in the presence and absence of CO, reaction condition: 463 K, 10 mg catalyst, 2 h, 20 mL H₂O, 3 bar O₂, 6 bar CO/or 6 bar Ar instead of CO, 0.175 mmol CH₃COOH.